# NEURAL OPTIMAL TRANSPORT FOR SUBSET ALIGNMENT

## ABSTRACT

We propose approaches for static and dynamic neural optimal transport with a relaxed Monge formulation to create optimal transport maps from a source distribution to an optimized distribution constrained to have an upper-bounded density ratio to the target distribution. In machine learning applications, this allows to learn the mappings between imbalanced datasets, such that one dataset can be mapped to a reweighted subset of a target dataset, with the reweighting governed by the density ratio constraint. The density ratio is constrained to lie in $[0, c]$ by the $f$-divergence associated with the indicator function for $[0, c]$, where $c$ denotes the maximum allowable upweighting factor. In the static case, neural networks are employed to parameterize the Monge map between source and selected subset of the target distribution and the dual function for the constraint. In the dynamic case, two networks are also employed: first neural network parametrizes the time dependent potential whose gradient defines the velocity field and terminal value enforces the density ratio constraint, while the second parametrizes the interpolation between the samples from source and optimized terminal distribution satisfying both the density ratio bound and the continuity equation. Since the terminal distribution in subset alignment need not be equal to the target distribution, which is distinct from prior work on dynamic neural optimal transport, we explore an efficient sampling scheme guided by the terminal potential. We apply both the static and dynamic formulations on domain translations problems, and demonstrate that the relaxed problem yields a more meaningful Monge map in cases where there is natural alignment between source and target distributions, but the distributions are imbalanced.

## 1 INTRODUCTION

Gaspard Monge proposed the original idea of optimal transport as mathematical model for the problem of minimum-cost transportation of dirt from source location to a destination Monge (1781). In more modern parlance, given probability measures, $\mu$ defined on compact set $\mathcal{X} \subseteq \mathbb{R}^d$, $\nu$ defined on compact set $\mathcal{Y} \subseteq \mathbb{R}^d$, and the bounded uniformly continuous cost $\mathsf{c}(\cdot, \cdot) : \mathcal{X} \times \mathcal{Y} \to \mathbb{R}$, Monge formulation of optimal transport is stated as

$$\mathcal{D}_{\text{Monge}}(\mu, \nu) = \inf_{T \in \mathcal{J}(\mathcal{X}, \mathcal{Y})} \int_{\mathcal{X}} \mathsf{c}\left(\boldsymbol{x}, T(\boldsymbol{x})\right) \mu(\boldsymbol{x}) d\boldsymbol{x} \tag{1}$$
$$\text{s.t. } T_{\#}\mu = \nu$$

where the set $\mathcal{J}(\mathcal{X}, \mathcal{Y})$ denotes the set of measurable maps between $\mathcal{X}$ and $\mathcal{Y}$. Monge formulation of the optimal transport problem requires that the transport map of $T$ to be a deterministic function. In order to satisfy the constraint in the Monge problem 1, the transport map $T$ must cover $\nu$ upto some $\nu$-null sets. Usually, the cost $\mathsf{c}$ is non-linearly dependent on the transportation map $T$, making the problem 1 very cumbersome and very difficult to solve (Santambrogio, 2015; Villani et al., 2009).

Recently, neural networks have been widely employed to solve optimal transport problems. Seguy et al. (2018) employed stochastic gradient-based approaches to estimate the optimal transport (Monge) map for large-scale data. In comparison, earlier work (Genevay et al., 2016) only minimized the optimal transport loss using stochastic gradient-based methods, or, as in well-known

Wasserstein-GAN (Arjovsky et al., 2017; Gulrajani et al., 2017), employed the Kantorovich-Rubinstein duality to minimize the Waserstein-1 loss function for generative modeling; however, the resulting generator is not trained to minimize distance as in the Monge formulation. Conversely, optimal transport maps can realize generative models (Daniels et al., 2021; Rout et al., 2022; Korotin et al., 2023b; Amos, 2023). For squared Euclidean transport cost, transport plans have been either directly parameterized using non-convex neural networks, (Rout et al., 2022; Korotin et al., 2023b) or obtained by amortizing the convex conjugate as gradients of convex functions parameterized by input convex neural networks (Amos et al., 2017; Makkuva et al., 2020; Korotin et al., 2021a; Amos, 2023; Vesseron & Cuturi, 2024). With recent developments in the development of flow matching (Lipman et al., 2023; Liu et al., 2023; Albergo & Vanden-Eijnden, 2023) as a state-of-the-art method for image generation, considerable recent efforts have been made to develop an efficient neural network-based framework for dynamic optimal transport for a variety of trajectory inference and generative modeling problems (Pooladian et al., 2024; Neklyudov et al., 2023; 2024b).

While distinct from generative modeling, the Monge map is a meaningful concept for the alignment of two real distributions (neither of which is noise) from slightly different domains, as in unsupervised domain adaptation. In these cases, distributional imbalance creates challenges (Wu et al., 2019). There has been substantial theoretical work on partial optimal transport (Figalli, 2010; Caffarelli & McCann, 2010; Chizat et al., 2018b;a) where two measures are not required to be of equal mass, and Wasserstein Fisher-Rao distance (Chizat et al., 2018a;b; Bauer et al., 2016) which allows for mass growth and destruction during the transfer process. Recent work on neural optimal transport in these cases (Gazdieva et al., 2023; Choi et al., 2023; Yang & Uhler, 2019). In this work, we formulate a relaxed version of optimal transport that creates a new distribution whose density ratio to the target distribution is bounded.

We propose static and dynamic neural optimal transport formulations, under the constraint density ratio constraint. To minimize the expected ground distance[1], the transported distribution can have a support that is subset of the target support. This can be interpreted as a reweighted target distribution with mass concentrated entirely on the selected subset. Our key contributions are as follows: we formulate both static and dynamic subset alignment problems by replacing the target marginal constraint with a penalty based on an $f$-divergence corresponding to the convex indicator function of the set $[0, c]$, where $c = 1$ recovers standard optimal transport; we leverage dual formulations of our problems using neural networks, in particular, we employ Benamou-Brenier formulation (see equation 22 in the appendix) along with the Lagrange multiplier method to obtain the dual form of dynamic subset selection; we show that the dual formulations in both the static and dynamic yield a potential function defined over the target support, whose sign effectively distinguishes points within the selected subset from those outside it; and we apply our framework to unpaired domain translation problems and use the potential function for PU-learning.

## 2 METHODOLOGY

### 2.1 STATIC SUPPORT SUBSET-SELECTION

The Kantorovich formulation (Kantorovich, 1942) for the optimal transport problem is

$$\mathcal{W}(\mu, \nu) = \inf_{\pi} \int_{\mathcal{X} \times \mathcal{Y}} \mathsf{c}(\boldsymbol{x}, \boldsymbol{y}) \pi(\boldsymbol{x}, \boldsymbol{y}) d\boldsymbol{x} d\boldsymbol{y}, \quad \text{s.t.} \int_{\mathcal{Y}} d\pi(\boldsymbol{x}, \boldsymbol{y}) = \mu(\boldsymbol{y}), \int_{\mathcal{X}} d\pi(\boldsymbol{x}, \boldsymbol{y}) = \nu(\boldsymbol{x}),$$

(2)

where $\pi$ is a density defined on $\mathcal{X} \times \mathcal{Y}$. Our formulation of static support subset-selection for optimal transport is derived from a relaxed problem where the constraint on the first marginal of the joint density $\pi$ is maintained, while the second marginal $\int_{\mathcal{X}} \pi(\boldsymbol{x}, \boldsymbol{y}) d\boldsymbol{x} = \tilde{\nu}(\boldsymbol{y})$ is allowed to vary from within a range $[0, c]$ of the target density $\nu$, such that $0 \leq \frac{\tilde{\nu}(\boldsymbol{y})}{\nu(\boldsymbol{y})} \leq c$. The density $\tilde{\nu}$ can be interpreted as a reweighted target density $\tilde{\nu}(\boldsymbol{y}) = \omega(\boldsymbol{y})\nu(\boldsymbol{y}), \quad 0 \leq \omega(\boldsymbol{y}) \leq c$, where portions of the support can be up-weighted while others are down-weighted or removed. The relaxed constraint is equivalent to a case of the partial optimal transport relaxations using $f$-divergences introduced by

---

[1]While we focus on the Euclidean distance, more general distances can be considered.

(Séjourné et al., 2023)

$$\inf_{\pi} \int_{\mathcal{X} \times \mathcal{Y}} \mathsf{c}(\boldsymbol{x}, \boldsymbol{y}) \pi(\boldsymbol{x}, \boldsymbol{y}) d\boldsymbol{x} d\boldsymbol{y} + \mathcal{D}_{\imath_{[a,b]}}(\tilde{\nu} \| \nu) \quad \text{s.t.} \int_{\mathcal{Y}} \pi(\boldsymbol{x}, \boldsymbol{y}) d\boldsymbol{y} = \mu(\boldsymbol{x}), \tag{3}$$

where $\mathcal{D}_{\imath_{[a,b]}}$ is the range divergence with $\imath_{[a,b]}$ being the convex indicator function

$$\imath_{[a,b]}(r) = \begin{cases} 0, & r \in [a,b] \\ +\infty, & \text{o.w.} \end{cases}, \quad \imath_{[a,b]}^{*}(t) = \sup_{u \in [a,b]} (u \cdot t) = \max(-at, bt), \tag{4}$$

and $\imath_{[a,b]}^{*}$ denotes its Legendre-Fenchel conjugate. Since the function $\imath_{[a,b]}$ is convex lower semi-continuous, therefore $\imath_{[a,b]} = \imath_{[a,b]}^{**}$, we can apply the variational form of the $f$-divergence Nguyen et al. (2010), (Polyanskiy & Wu, 2025, Theorem 7.26), exploited by $f$-GAN (Nowozin et al., 2016), leading to a form requiring only expected values

$$\mathcal{D}_{\varphi}(\tilde{\nu} \| \nu) = \int_{\mathcal{Y}} \imath_{[a,b]}(\frac{\tilde{\nu}}{\nu}(\boldsymbol{y})) \nu(\boldsymbol{y}) d\boldsymbol{y} = \sup_{\eta} \underbrace{\int_{\mathcal{Y}} \eta(\boldsymbol{y}) \tilde{\nu}(\boldsymbol{y}) d\boldsymbol{y}}_{\mathbb{E}_{\tilde{\boldsymbol{y}} \sim \tilde{\nu}}[\eta(\tilde{\boldsymbol{y}})]} - \underbrace{\int_{\mathcal{Y}} \imath_{[a,b]}^{*}(\eta(\boldsymbol{y})) \nu(\boldsymbol{y}) d\boldsymbol{y}}_{\mathbb{E}_{\boldsymbol{y} \sim \nu}[\imath_{[a,b]}^{*}(\eta(\boldsymbol{y}))]}. \tag{5}$$

To match 3, we focus on $a = 0$ and $b = c \geq 1$, such that $\imath_{[0,c]}^{*}(t) = c \cdot \max(0, t)$ and for compactness denote $\eta_{+}(\boldsymbol{y}) = \max(0, \eta(\boldsymbol{y}))$. Introducing $\psi$ as a measurable function to act as a Lagrange multiplier to enforce the constraint in 3 and combining with equation 5 yields the problem

$$\inf_{\pi} \sup_{\psi, \eta} \int (\mathsf{c}(\boldsymbol{x}, \boldsymbol{y}) + \eta(\boldsymbol{y}) - \psi(\boldsymbol{x})) \pi(\boldsymbol{x}, \boldsymbol{y}) d\boldsymbol{x} d\boldsymbol{y} + \int \psi(\boldsymbol{x}) \mu(\boldsymbol{x}) d\boldsymbol{x} - c \int \eta_{+}(\boldsymbol{y}) \nu(\boldsymbol{y}) d\boldsymbol{y}. \tag{6}$$

As described in App. A.1, since $\mathsf{c}$ is convex and lower semi-continuous, we interchange the $\inf_{\pi}$ and $\sup_{\eta}$ and apply what is known as the $\mathsf{c}$-transform of $-\eta(\boldsymbol{y})$ (Santambrogio, 2015; Villani et al., 2009) to obtain the dual problem with measurable map $T : \mathcal{X} \rightarrow \mathcal{Y}$

$$\sup_{\eta} \inf_{T} \underbrace{\int_{\mathcal{X}} (\mathsf{c}(\boldsymbol{x}, T(\boldsymbol{x})) + \eta(T(\boldsymbol{x}))) \mu(\boldsymbol{x}) d\boldsymbol{x}}_{\mathbb{E}_{\boldsymbol{x} \sim \mu}[\mathsf{c}(\boldsymbol{x}, T(\boldsymbol{x})) + \eta(T(\boldsymbol{x}))]} - c \underbrace{\int_{\mathcal{Y}} \eta_{+}(\boldsymbol{y}) \nu(\boldsymbol{y}) d\boldsymbol{y}}_{\mathbb{E}_{\boldsymbol{y} \sim \nu}[c \cdot \max(0, \eta(\boldsymbol{y}))]}, \tag{7}$$

For the computational implementation $T$ and $\eta$ are parameterized using neural networks with associated parameters $\theta_T$ and $\theta_{\eta}$ and expectations are estimated using samples from $\mu$ and $\nu$ as described in the Algorithm 1.

---

**Algorithm 1:** (`Static-Neural-SS`) Learning Algorithm for Static Subset Selection

**Inputs** : Source distribution $\mu$ and target distributions $\nu$, cost function $\mathsf{c}(\cdot, \cdot)$, reweighting bound $c$, neural networks $T(\cdot, \theta_T)$ and $\eta(\cdot, \theta_{\eta})$, batch size $N$, number of updates $n_T$ and $n_{\eta}$, and optimizers $\text{optim}_T$ and $\text{optim}_{\eta}$.

**Outputs** : Sample based neural estimate for transport map $T$

1   **for** *all learning iterations* **do**
2    **for** $n_T$ *update steps* **do**
3     sample $\{\boldsymbol{x}_i\}_{i=1}^{N} \sim \mu$ and $\{\boldsymbol{y}_j\}_{j=1}^{N} \sim \nu$
4     compute $grad_{\theta_T} = \nabla_{\theta_T} \frac{1}{N} \sum_{i=1}^{N} \left[ \mathsf{c}(\boldsymbol{x}_i, T(\boldsymbol{x}_i, \theta_T)) + \eta(T(\boldsymbol{x}_i, \theta_T), \theta_{\eta}) \right]$
5     use $grad_{\theta_T}$ to update $\theta_T$ with $\text{optim}_T$
6    **end**
7    **for** $n_{\eta}$ *update steps* **do**
8     sample $\{\boldsymbol{x}_i\}_{i=1}^{N} \sim \mu$ and $\{\boldsymbol{y}_j\}_{j=1}^{N} \sim \nu$
9     compute $grad_{\theta_{\eta}} = \nabla_{\theta_{\eta}} \frac{1}{N} \sum_{i=1}^{N} \left[ c \cdot \max(0, \eta(\boldsymbol{y}_j, \theta_{\eta})) - \eta(T(\boldsymbol{x}_i, \theta_T), \theta_{\eta}) \right]$
10    use $grad_{\theta_{\eta}}$ to update $\theta_{\eta}$ with $\text{optim}_{\eta}$
11    **end**
12 **end**

---

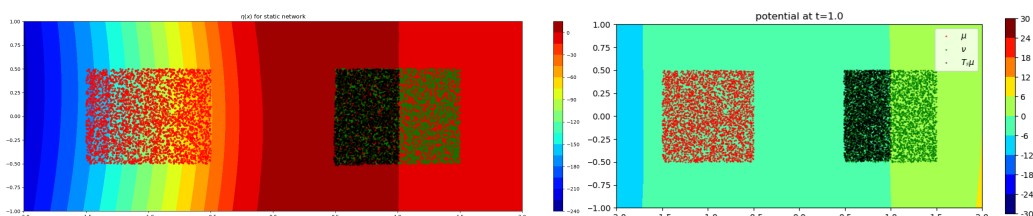

Figure 1: Subset Alignment between two uniform distributions in $\mathbb{R}^2$, (a) obtained by solving 7 and (b) obtained by solving 9 at $c = 2$ by using fully connected neural networks to parametrize $\eta$, $T$, $\varphi_t$ and $\rho_t$.

## 2.2 DYNAMIC SUBSET-SELECTION

Our formulation of dynamic support subset-selection for optimal transport is directly related to the Benamou-Brenier formulation of Wasserstein-2 distance. Similar to the static case, we replace the second marginal by a penalty based on the range divergence. The modified Benamou-Breneir problem is

$$\inf_{\rho_t, \boldsymbol{v}_t} \int_0^1 \int_\Omega \frac{\|\boldsymbol{v}_t(\boldsymbol{x})\|^2}{2} \rho_t(\boldsymbol{x}) d\boldsymbol{x}\, dt + \mathcal{D}_{\iota_{[0,c]}}(\rho_1 \| \nu)$$

$$\text{s.t.} \quad \frac{\partial}{\partial_t} \rho_t(\boldsymbol{x}) + \operatorname{div}(\rho_t(\boldsymbol{x}) \boldsymbol{v}_t(\boldsymbol{x})) = 0, \quad \rho_0(\boldsymbol{x}) = \mu(\boldsymbol{x}) \tag{8}$$

By introducing the Lagrange multiplier for $\varphi_t$ for continuity equation constraint, one can write the dual form of equation 8 as (see Appendix A.2 for details)

$$\sup_{\rho_t} \inf_{\varphi_t} \mathbb{E}_{\boldsymbol{x} \sim \mu}[\varphi_0(\boldsymbol{x})] + \mathbb{E}_{\boldsymbol{x} \sim \nu}[c \cdot \max(0, -\varphi_1(\boldsymbol{x}))] + \int_0^1 \mathbb{E}_{\boldsymbol{x}_t \sim \rho_t} \left[ \frac{\partial}{\partial t} \varphi_t(\boldsymbol{x}_t) + \frac{\|\nabla \varphi_t(\boldsymbol{x}_t)\|^2}{2} \right] dt. \tag{9}$$

From equation 41 and equation 9, one can see that, in addition to samples from source and target distributions, one additionally needs to have a mechanism to sample from an optimized distribution that interpolates between the source distribution and the terminal distribution that satisfies the range divergence to the target. This is essentially a generative modeling problem and the subject of many recent studies (Neklyudov et al., 2024a; Atanackovic et al., 2025; Du et al., 2024).

In flow-based models, instead of explicitly modeling $\rho_t$, samples $\boldsymbol{x}_0 \sim \mu$ and $\boldsymbol{x}_1 \sim \nu$ are used to generate $\boldsymbol{x}_t$ using an analytically defined interpolant Lipman et al. (2023); Liu et al. (2023); Albergo & Vanden-Eijnden (2023). In this work, we adapt the computational framework for learning Wasserstein-Lagrangian flows (WLF) (Neklyudov et al., 2024b) to parameterize $\rho_t$ in terms of $\mu$ and $\nu$. For a given $t \in [0, 1]$, WLF creates an interpolant $\boldsymbol{x}_t \sim \rho_t$ from $\boldsymbol{x}_0 \sim \mu$ and $\boldsymbol{x}_1 \sim \nu$ (independently sampled) as

$$\boldsymbol{x}_t = (1 - t)\boldsymbol{x}_0 + t\boldsymbol{x}_1 + t(1 - t) Q_t(\boldsymbol{x}_0, \boldsymbol{x}_1), \tag{10}$$

where $Q_t$ is time-dependent neural network, which internally uses an additional Heaviside step function input $t \geq 0.5$ (Neklyudov et al., 2024b). In the case when $c = 1$, subset alignment is equivalent to the optimal transport problem, therefore optimally $\rho_1^\star = \nu$, also given the optimal velocity field $\boldsymbol{v}_t^\star = \nabla \varphi_t^\star$, the optimal interpolant $\boldsymbol{x}_t^\star \sim \rho_t^\star$ is related to $\boldsymbol{v}_t^\star$ by

$$\boldsymbol{x}_t^\star = \begin{cases} \boldsymbol{x}_0 + \int_0^t \boldsymbol{v}_\tau^\star(\boldsymbol{x}_\tau) d\tau & t < 0.5 \\ \boldsymbol{x}_1 + \int_1^t \boldsymbol{v}_\tau^\star(\boldsymbol{x}_\tau) d\tau & t \geq 0.5 \end{cases}$$, resulting in forward integration from $\boldsymbol{x}_0$ for $t < 0.5$, and backward integration from $\boldsymbol{x}_1$ otherwise. However, for $c > 1$ $\rho_1^* \neq \nu$, therefore we can not directly draw samples $\boldsymbol{x} \sim \nu$ and propagate them backward for $t \geq 0.5$. Instead, an optimal interpolant could simply use the forward integration from $\boldsymbol{x}_0$. This means that $Q_t^\star$ would require the capacity to be a one-step integrator, which is not different from the $t < 0.5$ case for $c = 1$. However, in practice, the optimization of $\rho_t$ lags behind $\varphi_t$, and it may be advantageous to map samples from $\nu$ (or a distribution close to $\nu$) in order sample from $\rho_1$. We propose to sample $\tilde{\boldsymbol{x}}_1 \sim \tilde{\nu}$, where $\tilde{\nu}$ is chosen judiciously, and replace $\boldsymbol{x}_1$ with $\tilde{\boldsymbol{x}}_1$ in equation 10. In this case, the optimal interpolant $\boldsymbol{x}_t^\star \sim \rho_t^\star$ is still forward integration from $\boldsymbol{x}_0$ for $t < 0.5$, but for $t \geq 0.5$, $Q_t^\star(\boldsymbol{x}_0, \tilde{\boldsymbol{x}}_1)$ needs an internal map $S^\star$ such that the backward integration starts from a point sampled from the optimal

terminal distribution $\boldsymbol{x}_1^\star = S^\star(\tilde{\boldsymbol{x}}_1) \sim \rho_1^\star$ for $t \geq 0.5$, where $S_\#^\star \tilde{\nu} = \rho_1^\star$. If $\tilde{\nu} = \nu$ then $S^\star$ maps the original target to $\rho_1^\star$.

Our insight is to create $\tilde{\nu}$ by leveraging the fact that the optimal potential $\varphi_1^\star$ satisfies $\varphi_1^\star \leq 0$ almost surely on $\mathrm{supp}(\tilde{\nu})$ and $\varphi_1^\star > 0$ almost surely on $\mathrm{supp}(\nu) \setminus \mathrm{supp}(\tilde{\nu})$ (see Appendix B). Conditioning on the sign of $\varphi_1^\star$ allows us to sample from the selected subset of $\mathrm{supp}(\nu)$. Given a current estimate $\varphi_1$, we create $\nu_{\varphi_1}$, a distribution supported on the subset of the target where $\varphi_1 \leq 0$, as $\nu_{\varphi_1}(\boldsymbol{x}) = \nu(\boldsymbol{x} \mid \varphi_1(\boldsymbol{x}) \leq 0)$. When $\tilde{\nu} = \nu_{\varphi_1^\star} = \rho_1^\star$ then $S^\star(\tilde{x}_1) = \tilde{x}_1$. During training, however, $\varphi_1$ is suboptimal and may miss part of the support of the original target $\nu$, so we sample from the mixture $\alpha\nu_{\varphi_1} + (1-\alpha)\nu$, $\quad \alpha \in [0,1]$. Assuming $\varphi_1$ improves with training, we create a sequence of distributions, where at the $k$-th learning iteration, we can sample from the mixture

$$\tilde{\nu}^{(k)} = \alpha^{(k)} \nu_{\varphi_1^{(k)}} + (1 - \alpha^{(k)})\nu, \tag{11}$$

where $\alpha^{(k)}$ follows a monotonically non-decreasing scheduler with $\alpha^{(0)} = 0$ and $\alpha^{(\infty)} = 1$.[2] The complete procedure for solving the dynamic subset selection problem is outlined in Algorithm 2, wherein optimized parameters are $\theta_\varphi$ and $\theta_\rho$ (variables that are functions of parameters whose gradients are needed are explicitly noted).

---

**Algorithm 2:** (`Dynamic-Neural-SS`) Learning Algorithm for Dynamic Subset Selection

**Inputs** : Source distribution $\mu$ and target distributions $\nu$, time-dependent neural network $\varphi_t(\cdot, \theta_\varphi)$, network for the interpolant $Q_t(\cdot, \cdot, \theta_\rho)$ along with mixture schedule $\alpha^{(k)}$, batch size $N$, number of updates $n_\varphi$ and $n_\rho$, and optimizers $\mathrm{optim}_\varphi$ and $\mathrm{optim}_\rho$.

**Outputs** : Sample based neural estimate for $\varphi_t(\cdot, \theta_\varphi)$

1 **for** *learning iteration $k = 0, 1, \ldots$* **do**
2    **for** *$\varphi_t$ update steps* **do**
3      sample $\{\boldsymbol{x}_0^i\}_{i=1}^N \sim \mu$, $\{\boldsymbol{x}_1^i\}_{i=1}^N \sim \nu$, $\{\tilde{\boldsymbol{x}}_1^i\}_{i=1}^N \sim \tilde{\nu}^{(k)}$, and $\{t^i\}_{i=1}^N \sim \mathrm{Uniform}([0,1])$
4      compute $\tilde{\boldsymbol{x}}_t^i = (1-t^i)\boldsymbol{x}_0^i + t^i\tilde{\boldsymbol{x}}_1^i + t^i(1-t^i)Q_{t^i}(\boldsymbol{x}_0^i, \tilde{\boldsymbol{x}}_1^i, \theta_\rho), \quad \forall i \in \{1, \ldots, N\}$.
5      compute
$$grad_{\theta_\varphi} = \nabla_{\theta_\varphi} \frac{1}{N} \sum_{i=1}^N \left[ \frac{\partial}{\partial t}\varphi_{t^i}(\tilde{\boldsymbol{x}}_t^i, \theta_\varphi) + \frac{\|\nabla\varphi_{t^i}(\tilde{\boldsymbol{x}}_t^i, \theta_\varphi)\|^2}{2} \right.$$
$$\left. + \varphi_0(\boldsymbol{x}_0^i, \theta_\varphi) + c \cdot \max\left(0, -\varphi_1(\boldsymbol{x}_1^i, \theta_\varphi)\right) \right].$$
6      use $grad_{\theta_\varphi}$ to update $\theta_\varphi$ with $\mathrm{optim}_\varphi$.
7    **end**
8    **for** *$\rho_t$ update steps* **do**
9      sample $\{\boldsymbol{x}_0^i\}_{i=1}^N \sim \mu$, $\{\tilde{\boldsymbol{x}}_1^i\}_{i=1}^N \sim \tilde{\nu}_k$, $\{t^i\}_{i=1}^N \sim \mathrm{Uniform}([0,1])$.
10      compute $\tilde{\boldsymbol{x}}_t^i(\theta_\rho) = (1-t^i)\boldsymbol{x}_0^i + t^i\tilde{\boldsymbol{x}}_1^i + t^i(1-t^i)Q_{t^i}(\boldsymbol{x}_0^i, \tilde{\boldsymbol{x}}_1^i, \theta_\rho), \quad \forall i \in \{1, \ldots, N\}$.
11      compute
$$grad_{\theta_\rho} = \nabla_{\theta_\rho} \frac{1}{N} \sum_{i=1}^N \left[ \frac{\partial}{\partial t}\varphi_{t^i}(\tilde{\boldsymbol{x}}_t^i(\theta_\rho), \theta_\varphi) + \frac{\|\nabla\varphi_t(\tilde{\boldsymbol{x}}_t^i(\theta_\rho), \theta_\varphi)\|^2}{2} \right].$$
12      use $grad_{\theta_\rho}$ to update $\theta_\rho$ with $\mathrm{optim}_\rho$.
13    **end**
14 **end**

---

## 3 RELATED WORK

In addition to approaches mentioned in the introduction, we review advances in static neural optimal transport in the Appendix C.1. Our work on dynamic subset selection is most directly related to Lagrangian neural optimal transport (Pooladian et al., 2024), action-matching (Neklyudov et al., 2023) and Wasserstein Lagrangian flows (Neklyudov et al., 2024a). Pooladian et al. (2024). The neural optimal transport with Lagrangian costs framework (Pooladian et al., 2024) focuses on optimal

---

[2]Instead of trusting the sign directly, for small finite target datasets, we evaluate $\varphi_1$ for all $\boldsymbol{x}_1$ and retain the fraction $\frac{1}{c}$ of points with smallest value of $\varphi_1$ to obtain the sample from $\tilde{\nu}_\varphi$.

transport with different potentials in Euclidean space. Wasserstein-Lagrangian flows (Neklyudov et al., 2023) is mainly developed for the applications in cellular trajectory inference and quantum many body problems (Neklyudov et al., 2024b), and extends to more general settings on Wasserstein Fisher-Rao (Chizat et al., 2018a;b; Séjourné et al., 2023), with the ability to deal with mass growth/destruction, and different types of dynamics.

All these approaches and all flow-based models are developed for the cases when the marginals are to be preserved. (A more extensive review of recent work in dynamic neural optimal transport is included in Appendix C.2; additionally, since the optimal transport is intimately related to recent developments in generative modeling such as flow-matching and Schrödinger bridges, we also discuss the development in relation to optimal transport.) In contrast, with our proposed dynamic support subset-selection it is desirable to preserve one marginal and dynamically transfer that mass to the subset of the support of the other while minimizing the transport cost. Therefore our approach is an novel extension of prior work (Neklyudov et al., 2024a), and although we focused on the $\ell_2^2$ cost, our method is compatible with other Lagrangian costs (Pooladian et al., 2024), which could be useful for side-information as in semi-supervised domain adaptation.

## 4 EXPERIMENTS AND RESULTS

In this section we discuss the experimental results for susbet selection on an easily interpretable image-to-image case, where MNIST (Deng, 2012) is the source and EMNIST (Cohen et al., 2017) is the target. In this case, images of digits are a subset of the characters in EMNIST. We then apply our proposed approaches to domain translation on the FFHQ dataset (Karras et al., 2019) in 512-dimensional latent space of ALAE (Pidhorskyi et al., 2020).

### 4.1 MNIST → EMNIST DOMAIN TRANSLATION

MNIST data set contains 60,000 images of digits between 0-9 in training-subset and 10,000 images in test-subset. MNIST dataset is roughly balanced in the sense that the proportions of each data class in the dataset are roughly the same. EMNIST (byclass) dataset contains a set of English alphabet and numbers. EMNIST contains 62 imbalanced classes, of which 10 classes (between 0-9) represent numbers, and the rest of 52 classes represent upper and lower English case letters of the English alphabet. Roughly, 16% of EMNIST represent numbers and remaining 84% are alphabet.

Since our goal is to transfer MNIST images to EMNIST images such a way that MNIST digits are mapped to EMNIST digits while ignoring alphabet, we trained a neural network classifier to distinguish between digits and alphabet to evaluate the learned mapping (see implementation details in Appendix D.1. After training the classifier, we used both static and dynamic subset-selection approaches for domain translation between MNIST and EMNIST. Implementation details of the underlying models and there training are in Appendix D.2.

In our experiments, we trained and evaluated both static and dynamic models using both the static and dynamic subset-selection frameworks for $c \in \{1, 2, 4, 8\}$. For the dynamic case, similar to any flow based generative process, dynamic subset selection also requires a numerical integration (ODE integration with Euler type numerical integrator with 100 integration steps), but one-step integration can be used (Liu et al., 2023; 2024b). Figure 2 shows that perceptually, one-step integration performs worse in comparison to both static and ODE-based generation. We evaluated the classification

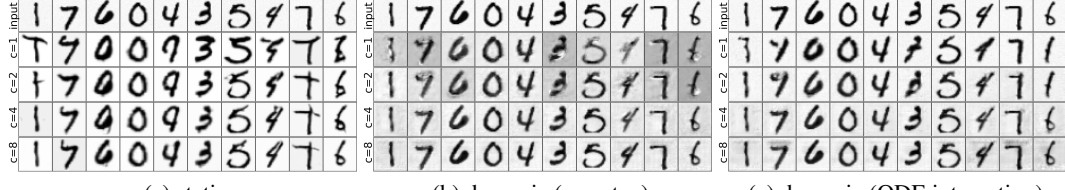

|  (a) static | (b) dynamic (one step) | (c) dynamic (ODE integration) |

Figure 2: Image translation outputs for MNIST →EMNIST

accuracy on translation of whole MNIST dataset, confusion matrices are given in Appendix E. A summary of accuracies of translated outputs are given in Table 1.

| Method | c=1 | c=2 | c=4 | c=8 |
|---|---|---|---|---|
| static | 46.93 | 75.33 | 82.44 | 87.32 |
| dynamic (one step) | 64.85 | 75.16 | 93.57 | 95.84 |
| dynamic (ODE) | 58.80 | 70.47 | 92.68 | 95.00 |

Table 1: Classification accuracies of translated images MNIST→EMNIST evaluated with using pretrained classifier.

## 4.2 POSITIVE-UNLABELED LEARNING

Positive Unlabeled (PU) learning is a binary classification problem in which only a subset of positive data is labeled, which is then used to train a model classifying between positive and negative data from an unlabeled (containing both positive and negative data) data set. PU Learning Bekker & Davis (2020); Kato et al. (2019); Chapel et al. (2020); Riaz et al. (2023). Since the sign of an optimal potential function in our framework differs between selected and unselected subsets, one can use it to distinguish between them positive and unlabeled datasets (see Appendix B for details). We applied applied both the static and dynamic optimal transport for PU learning on the 20 UCI-datasets (Kelly et al.) as in (Teisseyre et al., 2025), using the same settings with 75/25 train-test split on each dataset and the sampled completely randomly (SCAR) mechanism to selected and label points.

Networks were 5-layer MLPs with swish activation functions of appropriate input and output dimensions for both static and dynamic subset alignment with fixed learning learning rates for both static and dynamic models. Architecture and parameter details for each model are given in D.3. We trained 20 different models for each dataset using different train test splits, so in total we trained 400 models for static and 400 models for dynamic subset alignment. We adopted alternative sign and value based label assignment strategies for unlabeled dataset. Performance in terms of balanced accuracy for our approaches along with the top-performing baselines PUSB (Kato et al., 2019) and NTC-MI (Teisseyre et al., 2025) are given in Table 2.

| Dataset | $\pi$ | PUSB | NTC-MI | static | | dynamic | |
|---|---|---|---|---|---|---|---|
| | | | | sorted | sign | sorted | sign |
| Abalone | 0.16 | 0.544 ± 0.060 | 0.575 ± 0.025 | 0.561 ± 0.029 | 0.503 ± 0.008 | 0.555 ± 0.033 | 0.532 ± 0.030 |
| Banknote | 0.44 | 0.829 ± 0.050 | 0.922 ± 0.019 | 0.883 ± 0.037 | 0.882 ± 0.039 | 0.892 ± 0.048 | 0.895 ± 0.044 |
| Breast-w | 0.34 | 0.766 ± 0.145 | 0.870 ± 0.028 | 0.930 ± 0.028 | 0.941 ± 0.027 | 0.839 ± 0.197 | 0.831 ± 0.132 |
| Diabetes | 0.35 | 0.546 ± 0.042 | 0.700 ± 0.039 | 0.635 ± 0.044 | 0.635 ± 0.044 | 0.587 ± 0.094 | 0.603 ± 0.066 |
| Haberman | 0.26 | 0.513 ± 0.023 | 0.532 ± 0.066 | 0.539 ± 0.066 | 0.540 ± 0.067 | 0.528 ± 0.062 | 0.519 ± 0.070 |
| Heart | 0.44 | 0.527 ± 0.033 | 0.757 ± 0.053 | 0.637 ± 0.093 | 0.623 ± 0.089 | 0.508 ± 0.210 | 0.573 ± 0.139 |
| Ionosphere | 0.64 | 0.440 ± 0.085 | 0.755 ± 0.059 | 0.773 ± 0.091 | 0.762 ± 0.088 | 0.562 ± 0.215 | 0.602 ± 0.149 |
| Isolet | 0.04 | 0.793 ± 0.072 | 0.725 ± 0.006 | 0.881 ± 0.028 | 0.923 ± 0.030 | 0.673 ± 0.173 | 0.693 ± 0.202 |
| Jm1 | 0.19 | 0.628 ± 0.016 | 0.628 ± 0.013 | 0.576 ± 0.015 | 0.575 ± 0.010 | 0.573 ± 0.038 | 0.565 ± 0.026 |
| Kc1 | 0.15 | 0.645 ± 0.075 | 0.679 ± 0.030 | 0.604 ± 0.036 | 0.607 ± 0.035 | 0.611 ± 0.063 | 0.606 ± 0.054 |
| Madelon | 0.5 | 0.496 ± 0.030 | 0.519 ± 0.028 | 0.533 ± 0.025 | 0.523 ± 0.015 | 0.511 ± 0.027 | 0.505 ± 0.017 |
| Musk | 0.15 | 0.712 ± 0.036 | 0.767 ± 0.012 | 0.841 ± 0.018 | 0.847 ± 0.018 | 0.823 ± 0.020 | 0.840 ± 0.019 |
| Segment | 0.14 | 0.848 ± 0.074 | 0.803 ± 0.014 | 0.898 ± 0.038 | 0.927 ± 0.031 | 0.900 ± 0.042 | 0.935 ± 0.026 |
| Semeion | 0.1 | 0.569 ± 0.055 | 0.755 ± 0.022 | 0.824 ± 0.044 | 0.850 ± 0.067 | 0.699 ± 0.143 | 0.653 ± 0.144 |
| Sonar | 0.53 | 0.497 ± 0.041 | 0.573 ± 0.057 | 0.561 ± 0.091 | 0.524 ± 0.074 | 0.515 ± 0.107 | 0.511 ± 0.089 |
| Spambase | 0.39 | 0.821 ± 0.031 | 0.887 ± 0.014 | 0.786 ± 0.011 | 0.775 ± 0.010 | 0.703 ± 0.057 | 0.664 ± 0.067 |
| Vehicle | 0.26 | 0.549 ± 0.067 | 0.804 ± 0.042 | 0.806 ± 0.037 | 0.823 ± 0.032 | 0.639 ± 0.169 | 0.661 ± 0.152 |
| Waveform | 0.34 | 0.860 ± 0.012 | 0.829 ± 0.012 | 0.795 ± 0.015 | 0.743 ± 0.013 | 0.676 ± 0.071 | 0.551 ± 0.019 |
| Wdbc | 0.37 | 0.798 ± 0.155 | 0.801 ± 0.043 | 0.861 ± 0.068 | 0.845 ± 0.063 | 0.691 ± 0.211 | 0.641 ± 0.149 |
| Yeast | 0.31 | 0.517 ± 0.051 | 0.657 ± 0.024 | 0.630 ± 0.040 | 0.612 ± 0.049 | 0.590 ± 0.076 | 0.567 ± 0.063 |

Table 2: Comparison of average balanced accuracies of 20 models trained using static and dynamic subset alignment methods with PUSB and NTC-MI reported Teisseyre et al. (2025). Balanced accuracies for best performing methods are colored red and second best are colored blue.

## 4.3 FFHQ IMAGE TRANSLATION

We also apply our proposed approaches to the unpaired image translation problem. We followed the experimental setup of Gazdieva et al. (2024), where the FFHQ dataset embedded in the latent space of Adversarial Latent Autoencoder (ALAE) (Pidhorskyi et al., 2020), is divided either by gender

(man or woman) or age, as two orthogonal labels. Table 3 adapted from Gazdieva et al. (2024), shows the number of images for each class, where images with age $< 16$ are ignored, ages between 16 and 43 are labeled young, and the remainder are labeled old. Given these classes, the task is to learn to map a source distribution to a target distribution. There are four cases, young to old, old to young, man to woman, and woman to man. In order to evaluate the translation process, two classifiers pretrained in the ALAE latent space are used, one classifier is trained to classify young vs old and another to distinguish man vs woman. The target accuracy quantifies what proportion of translated images lie within the target-class boundary. The source accuracy quantifies whether the translated images retain the orthogonal label. For example, with young→old the source accuracy is whether the 'aged' image of a young source image retains the same gender.

Implementation details for both static and dynamic subset selection to the FFHQ dataset are given in Appendix D.4. Between young→old, old→young, man→woman and woman→man, it was observed that larger values of $c$ tend to preserve the source accuracy, but often have lower target accuracy. This can be related to the fact that for larger values of $c$, it takes more training steps to achieve the optimal subset selection.

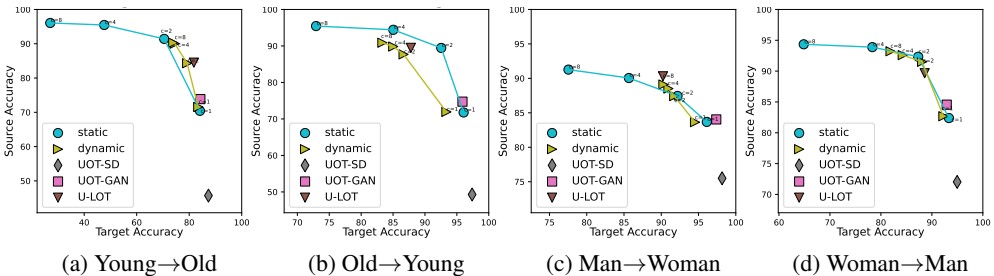

| (a) Young→Old | (b) Old→Young | (c) Man→Woman | (d) Woman→Man |
|---|---|---|---|

Figure 3: Accuracy curves for $c \in \{1, 2, 4, 8\}$, in comparison to results from LOT (Gazdieva et al., 2024), UOT-GAN (Yang & Uhler, 2019), and UOT-SD (Choi et al., 2024b).

We compared our methodology with Light Unbalanced optimal transport Gazdieva et al. (2024)(LOT), Yang & Uhler (2019)(UOT-GAN) and Choi et al. (2024b)(UOT-SD) and observed that methods which achieve better results in terms of target accuracy perform worse in terms of source class accuracy. This can be seem from Table 4 and Figure 3, using accuracy values reported by Gazdieva et al. (2024). Example translated images for static and dynamic are shown for old→young in Figure 4, with other cases provided in Appendix F.

| Class | Man | Woman |
|---|---|---|
| Young | 15K | 23K |
| Old | 7K | 3.5K |

Table 3: Division of FFHQ train images.

|  |  | c=1 | | c=2 | | c=4 | | c=8 | | UOT-SD | UOT-GAN | U-LOT |
|---|---|---|---|---|---|---|---|---|---|---|---|---|
| Task | Accuracy | static | dynamic | static | dynamic | static | dynamic | static | dynamic |  |  |  |
| Young→Old | Target | 84.09 | 83.45 | 70.47 | 79.23 | 47.63 | 74.51 | 27.07 | 73.93 | 87.33 | 84.25 | 81.78 |
|  | Class | 70.43 | 71.55 | 91.41 | 84.31 | 95.47 | 90.03 | 96.06 | 90.30 | 45.71 | 73.85 | 84.49 |
| Old→Young | Target | 96.06 | 93.36 | 92.55 | 86.65 | 85.04 | 85.03 | 72.93 | 83.33 | 97.39 | 95.88 | 87.79 |
|  | Class | 71.77 | 71.92 | 89.46 | 87.69 | 94.43 | 89.84 | 95.45 | 90.88 | 49.30 | 74.74 | 89.48 |
| Man→Woman | Target | 96.11 | 94.53 | 92.18 | 91.74 | 85.66 | 90.96 | 77.55 | 90.29 | 98.16 | 97.38 | 90.23 |
|  | Class | 83.68 | 83.64 | 87.45 | 87.43 | 90.05 | 88.52 | 91.27 | 89.11 | 75.50 | 84.04 | 90.30 |
| Woman→Man | Target | 93.34 | 92.26 | 87.32 | 88.09 | 78.32 | 84.28 | 64.86 | 81.89 | 94.96 | 92.91 | 88.59 |
|  | Class | 82.39 | 82.68 | 92.33 | 91.51 | 93.89 | 92.59 | 94.35 | 93.22 | 72.03 | 84.56 | 89.66 |

Table 4: Target and source accuracy (%) for different domain translations on the FFHQ dataset. Dynamic subset selection is evaluated using Euler integration with 100 steps.

## 5 DISCUSSION AND CONCLUSION

Practically, one important matter of concern for the utility of Wasserstein distances is the fact that sample estimators of Wasserstein distances are cursed by dimensionality (Weed & Bach, 2019;

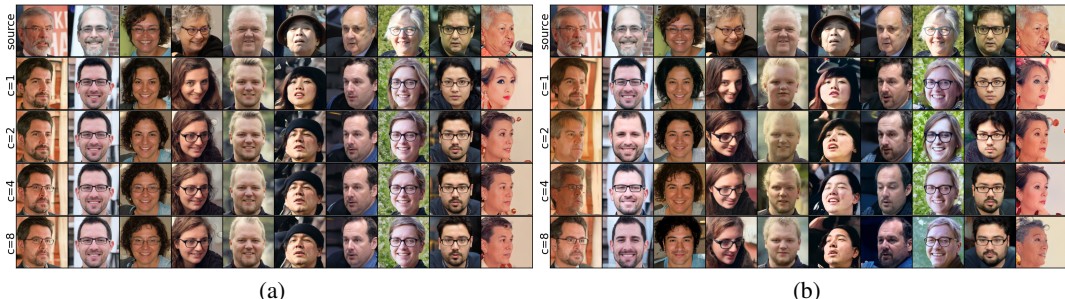

Figure 4: FFHQ old→young translation using (a) static and (b) dynamic subset selection. Dynamic subset selection is evaluated using Euler integration with 100 steps.

Fournier & Guillin, 2015), which can be alleviated to certain extent by employing the entropic regularization (Genevay et al., 2019; Feydy et al., 2019), which in the dynamic case is intimately connected with Schrödinger bridges.

Recently, unbalanced entropically-regularized optimal transport has been studied to model birth and death processes for population dynamics (Pariset et al., 2023; Neklyudov et al., 2023). Our approach can also be applied to model death processes, in cases where there is some canonical relationship between temporally ordered events, by treating $\mu$ as the final population of survivors and $\nu$ as the initial population.

Note the choice of $c$ is often critical in applications. While $c$ is interpretable, an automatic selection of $c$ based on the resulting transport cost, which was previously conducted for partial optimal transport in the discrete case (Phatak et al., 2023), may be possible. One consolidated approach would be to sample $c$ from a range and use multi-task learning for optimizing networks for varying $c$. In terms of implementation, this is possible using a scalar embedding of $c$ as used for embeddings of the time variables in dynamic networks. We would further like to point out that one can replace range divergence with more common divergences like KL divergence but we cannot use the sign of potential in that to distinguish between selected and rejected subsets.

Finally, we note that although we focused on relaxing the target distribution; the range-divergence framework could potentially be adapted to also relax the source distribution. A fully relaxed version may be applicable to other classes of problems.

In conclusion, our approaches for neural optimal transport with subset selection are motivated by problems that require translation between two distribution with reweighting and selection of the target. The results here, limited to image translation tasks on two datasets and 20 tabular PU-learning tasks, show that both a meaningful subset can be learned simultaneously with a Monge map. Unlike previous work, our dynamic formulation of allows for variation in the terminal distribution from the original target marginal, creating flows to the nearest subset.

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

## A PRELIMINARIES AND PROBLEM FORMULATION

Kantorovich Kantorovich (1942) reformulated the Monge problem by relaxing the constraint that supports of $\mu$ and $\nu$ should be related to each other by a functional relation $T$. Instead, he allowed $\mu$ and $\nu$ to be related to each other by a joint measure. Kantorovich's reformulation of the problem is a linear program and its solution exists for all convex lower-semi-continuous costs. Santambrogio (2015); Figalli & Glaudo (2023). The Kantorovich problem is,

$$
\mathcal{W}(\mu, \nu) = \inf_{\pi} \int_{\mathcal{X} \times \mathcal{Y}} \mathsf{c}(\boldsymbol{x}, \boldsymbol{y}) \pi(\boldsymbol{x}, \boldsymbol{y}) d\boldsymbol{x} d\boldsymbol{y}
$$
$$
\text{s.t.} \int_{\mathcal{Y}} d\pi(\boldsymbol{x}, \boldsymbol{y}) = \mu(\boldsymbol{y}), \quad \int_{\mathcal{X}} d\pi(\boldsymbol{x}, \boldsymbol{y}) = \nu(\boldsymbol{x}),
$$

(12)

where it can be observed that integrals $\int_{\mathcal{X}}$ and $\int_{\mathcal{Y}}$ marginalize with respect to spaces $\mathcal{Y}$ and $\mathcal{X}$, respectively. Therefore, $\pi$ must be the joint measure between $\mu$ and $\nu$ defined on the product space $\mathcal{X} \times \mathcal{Y}$, i.e. $\pi \in \mathcal{P}(\mathcal{X} \times \mathcal{Y})$. In other words, constraints in the Kantorovich problem ensure that every feasible $\pi$ must be joint distribution of $\mu$ and $\nu$. While in general, Kantorovich problem is much easier to solve in comparison to Monge problem, there are the conditions of practical importance where one can employ the solutions of Kantorovich problem to obtain the solution of Monge problem. Those conditions are more clearly discussed in terms of dual form of Kantorovich problem (Santambrogio, 2015), given as,

$$\mathcal{W}(\mu, \nu) = \sup_{f, g} \int_{\mathcal{X}} f(\boldsymbol{x})\mu(\boldsymbol{x})d\boldsymbol{x} + \int_{\mathcal{Y}} g(\boldsymbol{y})\mu(\boldsymbol{y})d\boldsymbol{y} \tag{13}$$
$$\text{s.t. } f(\boldsymbol{x}) + g(\boldsymbol{y}) \leq \mathsf{c}(\boldsymbol{x}, \boldsymbol{y})$$

The functions $f(\boldsymbol{x})$ and $g(\boldsymbol{y})$ are called Kantorovich potentials. By defining $\mathsf{c}$-conjugate (also called $\mathsf{c}$-transform) of $f(\boldsymbol{x})$, and $\bar{\mathsf{c}}$-conjugate (also called $\bar{\mathsf{c}}$-transform) of $g(\boldsymbol{y})$ as

$$f^{\mathsf{c}}(\boldsymbol{y}) = \inf_{\boldsymbol{x} \in \mathcal{X}} \mathsf{c}(\boldsymbol{x}, \boldsymbol{y}) - f(\boldsymbol{x}), \tag{14}$$

$$g^{\bar{\mathsf{c}}}(\boldsymbol{x}) = \inf_{\boldsymbol{x} \in \mathcal{X}} \mathsf{c}(\boldsymbol{x}, \boldsymbol{y}) - g(\boldsymbol{y}), \tag{15}$$

Using $\mathsf{c}$ and $\bar{\mathsf{c}}$ conjugates, Kantorovich problem is expressed as

$$\mathcal{W}(\mu, \nu) = \sup_{f} \int_{\mathcal{X}} f(\boldsymbol{x})\mu(\boldsymbol{x})d\boldsymbol{x} + \int_{\mathcal{Y}} f^{\mathsf{c}}(\boldsymbol{y})\nu(\boldsymbol{y})d\boldsymbol{y} \tag{16}$$

$$= \sup_{g} \int_{\mathcal{X}} g^{\bar{\mathsf{c}}}(\boldsymbol{x})\mu(\boldsymbol{x})d\boldsymbol{x} + \int_{\mathcal{Y}} g(\boldsymbol{y})\nu(\boldsymbol{y})d\boldsymbol{y} \tag{17}$$

Under very general conditions, one can relate the cost $\mathsf{c}$ with the support of optimal coupling solution $\pi^{\star}$ and optimal Kantorovich potentials $f^{\star}(\boldsymbol{x})$ and $f^{\mathsf{c}\star}(\boldsymbol{y})$ (Santambrogio, 2015, Theorem 1.37) by

$$\text{supp}(\pi^{\star}) \subset \{(\boldsymbol{x}, \boldsymbol{y}) \in \mathcal{X} \times \mathcal{Y} : f^{\star}(x) + f^{\star\mathsf{c}}(y) = \mathsf{c}(\boldsymbol{x}, \boldsymbol{y})\} \tag{18}$$

In discrete domains, above result is equivalent to Karush-Kuhn-Tucker (KKT) conditions for optimality. One can further relate the optimal solutions of Monge and Kantorovich problems using a landmark result by Gangbo & McCann (1996) , (Figalli & Glaudo, 2023, Theorem 2.7.1), which states that there exists an optimal Kantorovich coupling of the form $\pi^{\star} = (\text{Id} \times T^{\star})_{\#}\mu$, where $T^{\star}$ is Monge map satisfying

$$\nabla_{\boldsymbol{x}}\mathsf{c}(\boldsymbol{x}, T^{\star}(\boldsymbol{x})) + \nabla f^{\star}(x) = 0, \tag{19}$$

if the following conditions are satisfied

- $\mu$ is absolutely continuous,
- $\forall \boldsymbol{y} \in \mathcal{Y}$ the map $x \mapsto \mathsf{c}(\boldsymbol{x}, \boldsymbol{y})$ is differentiable, $\forall \boldsymbol{x} \in \mathcal{X}$,
- $\forall \boldsymbol{x} \in \mathcal{X}$ the gradient map $y \mapsto \nabla_{\boldsymbol{x}}\mathsf{c}(\boldsymbol{x}, \boldsymbol{y})$ is injective $\forall \boldsymbol{y} \in \mathcal{Y}$,
- and the gradient $\nabla_{\boldsymbol{x}}\mathsf{c}(\boldsymbol{x}, \boldsymbol{y})$ satisfies the local Lipschitz condition $\|\nabla_{\boldsymbol{x}}\mathsf{c}(\boldsymbol{x}, \boldsymbol{y})\| \leq C_r$ for all $\boldsymbol{x} \in \mathcal{B}_r$, where is $\mathcal{B}_r$ is ball of radius $r$ around $\boldsymbol{x}$.

When the cost can be written as $\mathsf{c}(\boldsymbol{x}, \boldsymbol{y}) = h(\boldsymbol{x} - \boldsymbol{y})$, where $h$ is strictly convex and translation invariant function, one can further relate the Monge mapping with optimal dual potential by (Santambrogio, 2015, Theorem 1.17)

$$T^{\star}(\boldsymbol{x}) = \boldsymbol{x} - \nabla h^{*} \circ \nabla f^{\star}(\boldsymbol{x}), \tag{20}$$

where $h^{*}$ is Legendre-Fenchel conjugate of $h$ given by $h^{*}(\boldsymbol{y}) = \sup_{x \in \mathcal{X}}\{\langle \boldsymbol{y}, \boldsymbol{x} \rangle - h(\boldsymbol{x})\}$. In the result above, when $h(\boldsymbol{x} - \boldsymbol{y}) = \frac{1}{2}\|\boldsymbol{x} - \boldsymbol{y}\|_2^2$, one obtains the result of celebrated Brenier theorem of optimal transport for squared-Euclidean costs with $T^{\star}(\boldsymbol{x}) = \nabla f^{\star}(\boldsymbol{x})$, where $f^{\star}(\boldsymbol{x})$ is convex (Brenier, 1991), (Figalli & Glaudo, 2023, Theorem 2.5.10) . The Brenier theorem on optimal transport differs from another important theorem on polar factorization (Brenier, 1991) stating that under very general conditions a square integrable vector field $v$ can be decomposed into the composition of gradient of a unique convex function $\xi$ and a unique measure-preserving map $u$, i.e. $v(\boldsymbol{x}) = \nabla \xi \circ u(\boldsymbol{x})$. Before

the discussion on dynamic formulation of the problem, we would like to point out that much of the recent work on static neural optimal transport rely on above results.

Benamou and Brenier formulated the Wasserstein distance with squared Euclidean cost as the kinetic energy minimization problem under the assumption that both the source $\mu$ and target $\nu$ distributions have finite second moments (Benamou & Brenier, 2000; Santambrogio, 2015; Figalli & Glaudo, 2023). Assuming that supports of both source and target distributions lie in a convex set $\Omega \subseteq \mathbb{R}^d$, whose normal at the boundary is given by $\mathbf{n} : \partial\Omega \to \mathbb{R}^d$, for a bounded and smooth velocity-field $\boldsymbol{v}_t(\boldsymbol{x}) : [0,1] \times \Omega \to \mathbb{R}^d$, such that $\langle \boldsymbol{v}_t(\boldsymbol{x}), \mathbf{n} \rangle|_{\partial\Omega} = 0$, the flow corresponding to $\boldsymbol{v}_t$ is given by

$$\frac{d}{dt}\Phi_t(\boldsymbol{x}_t) = \boldsymbol{v}_t(\Phi_t(\boldsymbol{x}_t)), \;\; \Phi_0(\boldsymbol{x}_0) = \boldsymbol{x}_0. \tag{21}$$

Considering that there also exists a probability path $\rho_t(\boldsymbol{x}) : [0,1] \times \Omega \to \mathbb{R}_+$, corresponding to the flow $\Phi_t(\boldsymbol{x})$ such that $\rho_t(\boldsymbol{x}) = \Phi_{t\#}\rho_0(\boldsymbol{x})$, Benamou-Brenier formulation of optimal transport is

$$\inf_{\rho_t, \boldsymbol{v}_t} \int_0^1 \int_\Omega \frac{\|\boldsymbol{v}_t(\boldsymbol{x})\|^2}{2} \rho_t(\boldsymbol{x}) \, d\boldsymbol{x} dt$$
$$\text{s.t. } \frac{\partial}{\partial_t}\rho_t(\boldsymbol{x}) + \operatorname{div}(\rho_t(\boldsymbol{x})\boldsymbol{v}_t(\boldsymbol{x})) = 0, \;\; \rho_0(\boldsymbol{x}) = \mu(\boldsymbol{x}), \;\; \rho_1(\boldsymbol{x}) = \nu(\boldsymbol{x}), \tag{22}$$

where $\operatorname{div}(\cdot)$ denotes divergence operator mapping scalar or vector fields to scalar, for the field $z_t(\boldsymbol{x})$ by $\operatorname{div}(z_t(\boldsymbol{x})) = \sum_i \frac{\partial}{\partial x_i} z_t(\boldsymbol{x})$. The optimal flow $\Phi_t^\star$ is related to Monge mapping $T^\star$ by displacement interpolation (McCann, 1997).

$$\Phi_t^\star = (1-t)\mathrm{Id} + tT^\star \tag{23}$$

It is important to mention that the Benamou-Brenier formulation can be extended to Wasserstein-$p$ distances, for $p > 1$, under the assumption that both source and target distributions have finite $p$-th moments (Santambrogio, 2015, chapters 5 & 6).

### A.1 Derivation of Static Neural Subset Selection

We denote the problem expressed in 6 as $\inf_\pi \sup_\psi \sup_\eta \mathcal{L}(\pi, \psi, \eta)$, where the Langrangian is

$$\mathcal{L}(\pi, \psi, \eta) = \int_{\mathcal{X} \times \mathcal{Y}} (\mathsf{c}(\boldsymbol{x}, \boldsymbol{y}) + \eta(\boldsymbol{y}) - \psi(\boldsymbol{x})) \pi(\boldsymbol{x}, \boldsymbol{y}) d\boldsymbol{x} d\boldsymbol{y} + \int_{\mathcal{X}} \psi(\boldsymbol{x}) \mu(\boldsymbol{x}) d\boldsymbol{x} \tag{24}$$

$$- c \int_{\mathcal{Y}} \eta_+(\boldsymbol{y})\nu(\boldsymbol{y})d\boldsymbol{y}. \tag{25}$$

We proceed to interchange the sup and inf,[3] which is allowed due to the strong duality property associated with optimal transport when the cost $\mathsf{c}$ is convex and lower semi-continuous,

$$\inf_\pi \sup_{\psi, \eta} \mathcal{L}(\pi, \psi, \eta) = \sup_{\eta, \psi} \inf_\pi \mathcal{L}(\pi, \psi, \eta). \tag{26}$$

Optimizing with respect to $\pi$ for given $\eta$ and $\psi$, the integrand in the first term of 24 is unbounded from below at any point $\mathsf{c}(\boldsymbol{x}, \boldsymbol{y}) + \eta(\boldsymbol{y}) - \psi(\boldsymbol{x}) < 0$. Thus, $\eta$ and $\psi$ need to ensure that $\mathsf{c}(\boldsymbol{x}, \boldsymbol{y}) + \eta(\boldsymbol{y}) - \psi(\boldsymbol{x}) \geq 0$. This constraint requires for any $\boldsymbol{x} \in \operatorname{supp}(\mu) \subseteq \mathcal{X}$, $\psi(\boldsymbol{x}) = \inf_{\boldsymbol{y} \in \mathcal{Y}}(\mathsf{c}(\boldsymbol{x}, \boldsymbol{y}) + \eta(\boldsymbol{y}))$ (Villani et al., 2009) (Theorem 5.10 and Remark 5.13). This definition of $\psi$ corresponds to the $\mathsf{c}$-transform of $-\eta(\boldsymbol{y})$ in the optimal transport literature (Santambrogio, 2015; Villani et al., 2009). Then, the inner infimum with respect to $\pi$ is attained with zero value if $\forall (\boldsymbol{x}, \boldsymbol{y}) \in (\mathcal{X} \times \mathcal{Y})$:

---

[3]This requires us to verify Slater's constraint qualifications, which are: (i) Primal is convex wrt $\pi$ ,(which is obvious), (ii) Dual is concave wrt $\eta$, which is also obvious (iii) relative interior for inequality constraints set is non-empty, which can be verified by looking at the fact that for any $\tilde{\nu}$ the distribution $\pi(\boldsymbol{x}, \boldsymbol{y}) = \mu(\boldsymbol{x})\tilde{\nu}(\boldsymbol{y})$ is feasible and one can see that if one defines the feasible set of coupling $\Pi_c(\mu, \nu) = \{\pi \in \mathcal{P}(\mathcal{X} \times \mathcal{Y}) : \pi_{\mathcal{X}} = \mu, \pi_{\mathcal{Y}} \leq c\nu\}$, then for $\forall 1 \leq c_0 \leq c_1, \Pi_{c_0}(\mu, \nu) \subseteq \Pi_{c_1}(\mu, \nu)$, which in other words mean that $\Pi_{c=1}(\mu, \nu)$ is a subset of feasible solutions for all values of $c > 1$, therefore relative-interior in non-empty.

$\pi(\boldsymbol{x}, \boldsymbol{y}) > 0 \implies \mathsf{c}(\boldsymbol{x}, \boldsymbol{y}) + \eta(\boldsymbol{y}) - \psi(\boldsymbol{x}) = 0$. Therefore, the dual problem becomes

$$\sup_{\eta} \int_{\mathcal{X}} \left( \inf_{\boldsymbol{y} \in \mathcal{Y}} (\mathsf{c}(\boldsymbol{x}, \boldsymbol{y}) + \eta(\boldsymbol{y})) \right) \mu(\boldsymbol{x}) d\boldsymbol{x} - c \int_{\mathcal{Y}} \eta_+(\boldsymbol{y}) \, \nu(\boldsymbol{y}) d\boldsymbol{y}. \tag{27}$$

$$= \sup_{\eta} \inf_{T} \int_{\mathcal{X}} \left( \mathsf{c}(\boldsymbol{x}, T(\boldsymbol{x})) + \eta(T(\boldsymbol{x})) \right) \mu(\boldsymbol{x}) d\boldsymbol{x} - c \int_{\mathcal{Y}} \eta_+(\boldsymbol{y}) \, \nu(\boldsymbol{y}) d\boldsymbol{y} \tag{28}$$

$$= \sup_{\eta} \inf_{T} \mathop{\mathbb{E}}_{\boldsymbol{x} \sim \mu} \left[ \mathsf{c}(\boldsymbol{x}, T(\boldsymbol{x})) + \eta(T(\boldsymbol{x})) \right] - c \mathop{\mathbb{E}}_{\boldsymbol{y} \sim \nu} [\eta_+(\boldsymbol{y})]. \tag{29}$$

$$= \inf_{T} \sup_{\eta} \mathop{\mathbb{E}}_{\boldsymbol{x} \sim \mu} \left[ \mathsf{c}(\boldsymbol{x}, T(\boldsymbol{x})) \right] + \mathop{\mathbb{E}}_{\boldsymbol{x} \sim \mu} \left[ \eta(T(\boldsymbol{x})) \right] - c \mathop{\mathbb{E}}_{\boldsymbol{y} \sim \nu} [\eta_+(\boldsymbol{y})]. \tag{30}$$

$$= \inf_{T} \mathop{\mathbb{E}}_{\boldsymbol{x} \sim \mu} \left[ \mathsf{c}(\boldsymbol{x}, T(\boldsymbol{x})) \right] + \mathcal{D}_{\imath_{[0,c]}}(T_\sharp \mu \| \nu) = \inf_{\substack{T \\ T_\# \mu \leq c\nu}} \mathop{\mathbb{E}}_{\boldsymbol{x} \sim \mu} \left[ \mathsf{c}(\boldsymbol{x}, T(\boldsymbol{x})) \right]. \tag{31}$$

Equation 27 and equation 28 are equal due to a theorem by (Rockafellar, 1976, Theorem 3A); equation 28 and equation 29 are equivalent by definition; equation 29 and equation 30 are equivalent since the function is convex with respect to $T$ and concave with respect to $\eta$; and equation 30 and equation 31 are equivalent by the variational formula for the $f$-divergence. Thus, we obtain a relaxed Monge formulation.

## A.2 DERIVATION OF DYNAMIC NEURAL SUBSET SELECTION

Combining the objective and constraints in 8 to obtain the Lagrangian

$$\mathcal{L}(\boldsymbol{v}_t, \rho_t, \psi_0, \varphi_t, \eta) = \overbrace{\int_0^1 \int_\Omega \frac{\|\boldsymbol{v}_t(\boldsymbol{x})\|^2}{2} \rho_t(\boldsymbol{x}) d\boldsymbol{x} dt}^{(\mathrm{I})} + \overbrace{\int_\Omega \psi_0(\boldsymbol{x}) \left( \rho_0(\boldsymbol{x}) - \mu(\boldsymbol{x}) \right) d\boldsymbol{x}}^{(\mathrm{II})}$$

$$+ \overbrace{\int_0^1 \int_\Omega \varphi_t(\boldsymbol{x}) \frac{\partial}{\partial_t} \rho_t(\boldsymbol{x}) d\boldsymbol{x} dt}^{(\mathrm{III})} + \overbrace{\int_0^1 \int_\Omega \varphi_t(\boldsymbol{x}) \mathrm{div}\left( \rho_t(\boldsymbol{x}) v_t(\boldsymbol{x}) \right) d\boldsymbol{x} dt}^{(\mathrm{IV})}$$

$$+ \overbrace{\sup_{\eta} \left( \int_\Omega \eta(\boldsymbol{x}) \rho_1(\boldsymbol{x}) d\boldsymbol{x} - c \int_\Omega \max(0, \eta(\boldsymbol{x})) \nu(\boldsymbol{x}) d\boldsymbol{x} \right)}^{(\mathrm{V})}. \tag{32}$$

Since $\rho_t(\boldsymbol{x})$ is supported on bounded subset $\Omega \subset \mathbb{R}^d$, one can change the order of integration. Therefore, for term (III), by changing the order of integration and then computing integration by parts one obtains,

$$(\mathrm{III}) = \int_\Omega \int_0^1 \varphi_t(\boldsymbol{x}) \frac{\partial}{\partial_t} \rho_t(\boldsymbol{x}) d\boldsymbol{x} dt = \int_\Omega \varphi_1(\boldsymbol{x}) \rho_1(\boldsymbol{x}) d\boldsymbol{x} - \int_\Omega \varphi_0(\boldsymbol{x}) \rho_0(\boldsymbol{x}) d\boldsymbol{x}$$

$$- \int_0^1 \int_\Omega \frac{\partial}{\partial_t} \varphi_t(\boldsymbol{x}) \rho_t(\boldsymbol{x}) d\boldsymbol{x} dt. \tag{33}$$

In order to simplify (IV), we can use product rule of derivatives to write

$$\varphi_t(\boldsymbol{x}) \mathrm{div}(\rho_t(\boldsymbol{x}) \boldsymbol{v}_t(\boldsymbol{x})) = \mathrm{div}\left( \varphi_t(\boldsymbol{x}) \rho_t(\boldsymbol{x}) \boldsymbol{v}_t(\boldsymbol{x}) \right) - \rho_t(\boldsymbol{x}) \langle \nabla \varphi_t(\boldsymbol{x}), \boldsymbol{v}_t(\boldsymbol{x}) \rangle.$$

Therefore by combining above identity with Gauss's theorem one obtains

$$(\mathrm{IV}) = \int_0^1 \int_\Omega \mathrm{div}\left( \varphi_t(\boldsymbol{x}) \rho_t(\boldsymbol{x}) \boldsymbol{v}_t(\boldsymbol{x}) \right) d\boldsymbol{x} dt - \int_0^1 \int_\Omega \rho_t(\boldsymbol{x}) \langle \nabla \varphi_t(\boldsymbol{x}), \boldsymbol{v}_t(\boldsymbol{x}) \rangle d\boldsymbol{x} dt$$

$$= \underbrace{\int_0^1 \oint_{\partial\Omega} \varphi_t(\boldsymbol{x}_t) \rho_t(\boldsymbol{x}_t) \langle \boldsymbol{v}_t(\boldsymbol{x}), d\boldsymbol{n} \rangle dt}_{=0} - \int_0^1 \int_\Omega \rho_t(\boldsymbol{x}) \langle \nabla \varphi_t(\boldsymbol{x}), \boldsymbol{v}_t(\boldsymbol{x}) \rangle d\boldsymbol{x} dt. \tag{34}$$

From the boundary condition on optimal transport (see the discussion above Equation 21, also Figalli & Glaudo (2023)-section 4.1) , the first part of the right-hand side of 35 is zero; therefore,

$$(\mathrm{IV}) = \int_0^1 \int_\Omega \varphi_t(\boldsymbol{x}) \cdot \mathrm{div}(\rho_t(\boldsymbol{x}) \boldsymbol{v}_t(\boldsymbol{x})) d\boldsymbol{x} dt = - \int_0^1 \int_\Omega \langle \nabla \varphi_t(\boldsymbol{x}), \boldsymbol{v}_t(\boldsymbol{x}) \rangle \rho_t(\boldsymbol{x}) d\boldsymbol{x} dt. \tag{35}$$

In order to eliminate primal variable $v_t(x)$, substitute Equation 33 and Equation 35 into 32 and compute the variational-derivative to obtain the stationary condition. For that one can write the terms of Lagrangian depending on $v_t(x)$ as

$$\tilde{\mathcal{L}}(v_t) = \int_0^1 \int_\Omega \left( \frac{\|v_t(x)\|^2}{2} - \langle \nabla\varphi_t(x), v_t(x) \rangle \right) \rho_t(x) dx dt. \tag{36}$$

With the additive perturbation function $\tau_t$ vanishing at $t = 0$ and $t = 1$ and a scalar $\varepsilon$, the Lagrangian $\tilde{\mathcal{L}}(v_t + \varepsilon\tau_t)$ is

$$\tilde{\mathcal{L}}(v_t + \varepsilon\tau_t) = \int_0^1 \int_\Omega \left( \frac{\|v_t(x) + \varepsilon\tau_t(x)\|^2}{2} - \langle \nabla\varphi_t(x), \ v_t(x) + \varepsilon\tau_t(x) \rangle \right) \rho_t(x) dx dt$$

$$= \int_0^1 \int_\Omega \left( \frac{\|v_t(x)\|^2}{2} - \langle \nabla\varphi_t(x), v_t(x) \rangle \right) \rho_t(x) dx dt$$

$$+ \int_0^1 \int_\Omega \left( \varepsilon^2 \frac{\|\tau_t(x)\|^2}{2} + \varepsilon\langle v_t(x) - \nabla\varphi_t(x), \ \tau_t(x) \rangle \right) \rho_t(x) dx dt,$$

and variational derivative is

$$\delta\tilde{\mathcal{L}}(v_t(x))\Big|_{v_t} = \frac{d}{d\varepsilon}\Big|_{\varepsilon=0} \tilde{\mathcal{L}}(v_t + \varepsilon\tau_t) = \int_0^1 \int_\Omega \langle v_t(x) - \nabla\varphi_t(x), \ \tau_t(x) \rangle \rho_t(x) dx dt. \tag{37}$$

The stationarity condition requires $\delta_{v_t}\tilde{\mathcal{L}}(v_t(x)) = 0$. For arbitrary perturbation $\tau_t(x)$, the variation $\delta_{v_t}\tilde{\mathcal{L}}(v_t(x)) = 0$ if and only if

$$v_t(x) = \nabla\varphi_t(x). \tag{38}$$

Therefore one can write the Lagrangian as

$$\mathcal{L}(\rho_t, \psi_0, \varphi_t, \eta) = \int_\Omega \psi_0(x) \cdot (\rho_0(x) - \mu(x)) \, dx + \int_\Omega \varphi_1(x)\rho_1(x) dx - \int_\Omega \varphi_0(x)\rho_0(x) dx$$

$$+ \int_\Omega \eta(x)\rho_1(x) dx - c \cdot \int_\Omega \max(0, \eta(x))\nu(x) dx$$

$$- \int_0^1 \int_\Omega \left( \frac{\partial}{\partial t}\varphi_t(x) + \frac{\|\nabla\varphi_t(x)\|^2}{2} \right) \rho_t(x) dx dt.$$

Similarly, by computing $\delta_{\psi_0}\mathcal{L}$ and $\delta_{\rho_1}\mathcal{L}$ using stationary conditions, one obtains the condition,

$$\psi_0(x) = \varphi_0(x), \tag{39}$$
$$\eta(x) = -\varphi_1(x). \tag{40}$$

Therefore the Lagrangian is simplified to

$$\mathcal{L}(\rho_t, \varphi_t, \eta) = -\int_\Omega \varphi_0(x)\mu(x) dx - c\int_\Omega \max(0, -\varphi_1(x))\nu(x) dx$$

$$- \int_0^1 \int_\Omega \left( \frac{\partial}{\partial t}\varphi_t(x) + \frac{\|\nabla\varphi_t(x)\|^2}{2} \right) \rho_t(x) dx dt. \tag{41}$$

The simplified problem (equation 9 in the main body) is

$$\sup_{\rho_t} \inf_{\varphi_t} \mathbb{E}_{x\sim\mu}\left[\varphi_0(x)\right] + c \cdot \mathbb{E}_{x\sim\nu}\left[\max(0, -\varphi_1(x))\right] + \int_0^1 \mathbb{E}_{x_t\sim\rho_t}\left[ \frac{\partial}{\partial t}\varphi_t(x_t) + \frac{\|\nabla\varphi_t(x_t)\|^2}{2} \right] dt.$$

## B THRESHOLDING FOR PU-LEARNING AND REJECTION SAMPLING

Our idea of rejection sampling and thresholding for PU-Learning is based on the fact that the dual form of range divergence is zero when the supremum in the dual is attained by the function $\eta^\star(x)$ with $\tilde{\nu}(x) = \rho_1^\star(x)$ i.e.

$$\mathbb{E}_{x\sim\tilde{\nu}}[\eta^\star(x)] - c \mathbb{E}_{x\sim\nu}[\texttt{ReLU}(\eta^\star(x))] = 0 \tag{42}$$

By defining $\mathcal{A} = \mathrm{supp}(\nu)$, $\tilde{\mathcal{A}} = \mathrm{supp}(\tilde{\nu})$, and $\bar{\mathcal{A}} = \mathcal{A}/\tilde{\mathcal{A}}$, one can also see that $\tilde{\mathcal{A}} \cap \bar{\mathcal{A}} = \emptyset$, therefore one can write Equation 42 as

$$\int_{\tilde{\mathcal{A}}} \eta^\star(\boldsymbol{x}) \tilde{\nu}(\boldsymbol{x}) d\boldsymbol{x} - c \int_{\tilde{\mathcal{A}}} \mathtt{ReLU}\left(\eta^\star(\boldsymbol{x})\right) \nu(\boldsymbol{x}) d\boldsymbol{x} - c \int_{\bar{\mathcal{A}}} \mathtt{ReLU}(\eta^\star(\boldsymbol{x})) \nu(\boldsymbol{x}) d\boldsymbol{x} = 0. \quad (43)$$

One can further write

$$\eta^\star(\boldsymbol{x}) = \mathtt{ReLU}\left(\eta^\star(\boldsymbol{x})\right) - \mathtt{ReLU}\left(-\eta^\star(\boldsymbol{x})\right). \quad (44)$$

After substituting Equation 44 into Equation 43 one obtains

$$\overbrace{\int_{\tilde{\mathcal{A}}} \mathtt{ReLU}(\eta^\star(\boldsymbol{x}))(\tilde{\nu}(\boldsymbol{x}) - c\nu(\boldsymbol{x})) d\boldsymbol{x}}^{\text{LHS}} = \overbrace{\int_{\tilde{\mathcal{A}}} \mathtt{ReLU}(-\eta^\star(\boldsymbol{x})) \tilde{\nu}(\boldsymbol{x}) d\boldsymbol{x} + c \int_{\bar{\mathcal{A}}} \mathtt{ReLU}(\eta^\star(\boldsymbol{x})) \nu(\boldsymbol{x}) d\boldsymbol{x}}^{\text{RHS}}$$
$$(45)$$

The dual form Equation 43 is optimal with zero duality gap, if the primal form satisfies $\forall \boldsymbol{x} \in \mathcal{A}$, $\iota_{[0,c]}\left(\frac{\tilde{\nu}}{\nu}(\boldsymbol{x})\right) = 0$, which can also be restricted to $\forall \boldsymbol{x} \in \tilde{\mathcal{A}} \; \iota_{[0,c]}\left(\frac{\tilde{\nu}}{\nu}(\boldsymbol{x})\right) = 0$. This is equivalent to $\tilde{\nu}(\boldsymbol{x}) \leq c\nu(\boldsymbol{x})$ almost-everywhere in $\tilde{\mathcal{A}}$. Therefore, one can say that $0 \geq \text{LHS}$ and also

$$0 \geq \overbrace{\int_{\tilde{\mathcal{A}}} \mathtt{ReLU}(-\eta^\star(\boldsymbol{x})) \nu(\boldsymbol{x}) d\boldsymbol{x} + c \int_{\bar{\mathcal{A}}} \mathtt{ReLU}(\eta^\star(\boldsymbol{x})) \nu(\boldsymbol{x}) d\boldsymbol{x}}^{\text{RHS}} \quad (46)$$

We can now see that both integrands in Equation 46 are nonnegative and sum to a value less than or equal to zero, which is only possible if both are equal to zero. Therefore, one can write

$$0 = \int_{\tilde{\mathcal{A}}} \mathtt{ReLU}(-\eta^\star(\boldsymbol{x})) \nu(\boldsymbol{x}) d\boldsymbol{x} + c \int_{\bar{\mathcal{A}}} \mathtt{ReLU}(\eta^\star(\boldsymbol{x})) \nu(\boldsymbol{x}) d\boldsymbol{x} \quad (47)$$

Further, two non-negative integrals are evaluated on two mutually exclusive sets, therefore to have sum equal to zero value we can conclude that each integral is zero individually. Therefore, we can write

$$0 = \int_{\tilde{\mathcal{A}}} \mathtt{ReLU}(-\eta^\star(\boldsymbol{x})) \nu(\boldsymbol{x}) d\boldsymbol{x} = c \int_{\bar{\mathcal{A}}} \mathtt{ReLU}(\eta^\star(\boldsymbol{x})) \nu(\boldsymbol{x}) d\boldsymbol{x} \quad (48)$$

The Equation 48 is therefore equivalent to following element-wise test

$$\begin{aligned} \eta^\star(\boldsymbol{x}) &\geq 0, \text{ almost surely in } \tilde{\mathcal{A}} \\ \eta^\star(\boldsymbol{x}) &< 0, \text{ almost surely in } \bar{\mathcal{A}} \end{aligned} \quad (49)$$

Additionally, from the Equation 47, one can also conclude that

$$\overbrace{\int_{\tilde{\mathcal{A}}} \mathtt{ReLU}(\eta^\star(\boldsymbol{x}))(\tilde{\nu}(\boldsymbol{x}) - c\nu(\boldsymbol{x})) d\boldsymbol{x}}^{\text{LHS}} = 0, \quad (50)$$

which is a complementary slackness condition in the sense that $\tilde{\nu}(\boldsymbol{x}) < c\nu(\boldsymbol{x}) \implies \eta^*(\boldsymbol{x}) = 0$ almost every-where in $\mathcal{A}$. During the neural network training with finite data-points, potential function $\eta$ is usually suboptimal and its sign cannot be relied, therefore instead of directly using the sign, one can sort values of potential at data points and select predetermined proportion (prior) of data-points. Therefore for training PU-learning models, we applied both sign and sorting based filtration of data. Form the figures 1a and 1b, one can observed that for the optimal potential for static problem exactly follows equation 49, whereas in the dynamic case the sign of $\varphi_1$ is inverted, which is due to the relation obtained in equation 40, which ensures that for the at optimal $\varphi_1^\star$ following relation holds

$$\begin{aligned} \varphi_1^\star(\boldsymbol{x}) &\leq 0, \text{ almost surely in } \tilde{\mathcal{A}} \\ \varphi_1^\star(\boldsymbol{x}) &> 0, \text{ almost surely in } \bar{\mathcal{A}}. \end{aligned} \quad (51)$$

The Figure 5 gives snapshots of the transition of $\varphi_t(\boldsymbol{x})$ between $t = 0$ and $t = 1$ for the dynamic subset alignment results shown 1b.

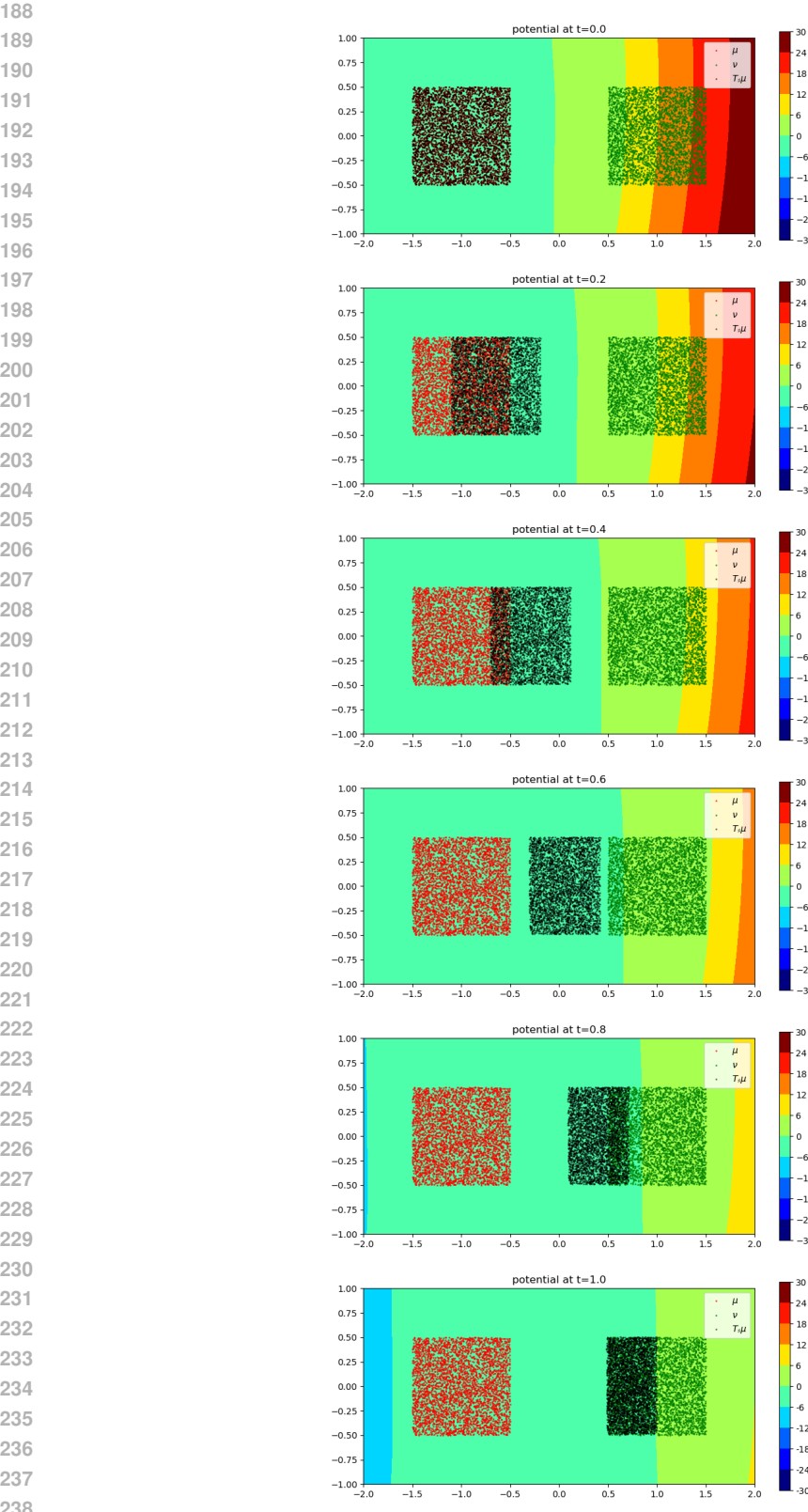

Figure 5: $\varphi_t$ between $t = 0$ and $t = 1$ for subset alignment between 2D uniform distributions for which $\varphi_1$ is also shown in Figure 1b, It can be seen that unlike $\eta$ in static problem $\varphi_t$ is function of time and varies with $t$.

## C   Survey of Recent Work on Neural Optimal Transport

In this section, we discuss the recent related work on computational optimal transport and its applications. More specifically, we consider the works which are related to neural estimation of optimal transport maps with occasional reference to theoretical developments.

### C.1   Static Neural Optimal Transport

Seguy et al. (2018) employed stochastic gradient based approaches in one of the earliest works to estimate the optimal transport map using neural networks. Notably, the work by Seguy et al. (2018) differed from early work Genevay et al. (2016) in the sense that the later work employed stochastic gradient based methods to estimate the transport plan for large scale data, whereas earlier work Genevay et al. (2016) only minimized the optimal transport loss using stochastic-gradient based methods. This is also in contrast to the well-known Wasserstein-GAN Arjovsky et al. (2017); Gulrajani et al. (2017) that employs the Kantorovich-Rubinstein duality to minimize the Wasserstein-1 loss function for generative modeling, where neural networks are employed as parameterizations for both dual-potential and data generator, but do not provide transport plans. Finally, the Sinkhorn-GAN employs an approximation of the discrete Wasserstein distance between latent representations of data and that of samples from non-informative prior Genevay et al. (2018) for generative modeling. Now, we can see clear distinction between two different classes of approaches employing Wasserstein distances in generative modeling, the first class of works concerns with employing Wasserstein distance as a loss for generative modeling, without any explicit concern for obtaining the underlying transport plan across the distributions Arjovsky et al. (2017); Gulrajani et al. (2017). The second class seeks to learn a transport plan to realize the generative model.

Efforts to learn Monge maps were motivated by a theorem by Brenier (1991), which essentially states that, for continuous distributions with squared-Euclidean transportation cost, the optimal solution of the Monge problem is the gradient of a convex function (Figalli & Glaudo, 2023, (Theorem 2.5.10)). Therefore initially, gradient of input-convex neural networks (ICNN) Amos et al. (2017) we employed to estimate the transport plan for the Wasserstein-2 distance Makkuva et al. (2020); Korotin et al. (2021a;b). This approach has also been employed to supervised conditional neural Monge maps (Bunne et al., 2022a) and unbalanced optimal transport (Lübeck et al., 2022). The study by Amos et al. (2023) focuses on the development of an efficient neural optimal solution that could be implemented quickly in more practical scenarios. This approach to solve Wasserstein-2 distances employing convex potentials involves computationally challenging evaluation of the Fenchel conjugate of a ICNN parameterized convex function. More recent work in this direction focuses on improved optimization strategies and better ICNN architectures to bypass problems related to Fenchel conjugate evaluations and ICNN training Amos (2023); Vesseron & Cuturi (2024). Recent work also focuses on some batch-based schemes have also been devised to improve the regularity of learned neural Monge maps Uscidda & Cuturi (2023); Eyring et al. (2024).

Another recent direction of work is based on the idea that ICNNs can be overly restrictive, therefore more general neural network architectures should be employed to directly parameterize the transport maps Rout et al. (2022); Korotin et al. (2023b). The work by Fan et al. (2022a; 2023) focuses on employing neural networks to approximate the solution for Monge's transport problem also draws inspiration from the recent developments in neural-network-based parametric realizations for approximating Kantorovich plans. Recently neural optimal transport has also been extended to unbalanced transportation setting (Yang & Uhler, 2019; Choi et al., 2023). Another work directly related to static subset selection problem is (Gazdieva et al., 2023).

Unless there is a corresponding Monge mapping (Choi et al., 2024a; Mokrov et al., 2024; Geuter et al., 2025), optimal transport requires a stochastic transport plans. A recent body of work (Korotin et al., 2023b;a; Asadulaev et al., 2024) deals with learning transportation plans using a weaker formulation of optimal transport (Gozlan et al., 2017; Backhoff-Veraguas et al., 2019) along with noise outsourcing techniques, which is also extended to more general costs. Apart from the applications in image translation (Korotin et al., 2023b), neural optimal transport has been applied for bio-medical image registration (Kim et al., 2024) and to study single cell perturbations (Bunne et al., 2023). Neural optimal transport have also been employed for metric learning (Howard et al., 2024; Scarvelis & Solomon, 2023).

## C.2 Dynamic Neural Optimal Transport

The potential applications of dynamic optimal transport in the cellular trajectory inference (Tong et al., 2020) and its connections with flow based models for generative modeling (Huang et al., 2021; Huguet et al., 2022) has been instrumental in the recent research developments in this direction. Jordan-Kinderlehrer-Otto flow (JKO) is time discretization scheme to solve Wasserstein gradient flows for different energy functionals (Jordan et al., 1998; Santambrogio, 2017). Therefore, a lot of effort done in that regard is focused on neural network parameterized schemes to solve JKO-flow problem for both cellular trajectory inference and generative modeling (Ma et al., 2021; Fan et al., 2022b; Lambert et al., 2022; Bunne et al., 2022b; Xu et al., 2023; Choi et al., 2023; 2024c; Altekrüger et al., 2023; Mokrov et al., 2021; Alvarez-Melis et al., 2022). JKO-scheme has also been studied for the applications related to molecular discovery (Alvarez-Melis et al., 2022). A recent study deals with convergence properties of JKO-based generative models (Cheng et al., 2024).

Recent developments in flow-matching models based on flow matching (Lipman et al., 2023; Albergo & Vanden-Eijnden, 2023; Liu et al., 2023) for generative modeling lead to even more interest in the development of algorithms to solve dynamic optimal transportation problems. Action-Matching based framework lead to the development of a more general framework to solve both trajectory inference and generative modeling problems (Neklyudov et al., 2023) for the cases where one could also sample from the trajectory between two terminal marginals. Rectified flow-matching (Liu et al., 2023; 2024b) uses the neural-optimal transport in additional rectification step to improve the linearity of flows, so that after training the model, images could be generated efficiently with only a single-step integration along straight lines paths. For generative modeling, in contrast to target-conditional flow matching (Lipman et al., 2023), where during training, flows are conditioned on target samples, discrete optimal transport conditioned flow-matching employs the mini-batch optimal transport to create the conditionals (Pooladian et al., 2023; 2024; Tong et al., 2024b). Another recent work (Kornilov et al., 2024) attempts to alleviate the error accumulation problems associated with mini-batch optimal transport by learning straight paths between source and target distributions in single step. Flow-matching (Albergo & Vanden-Eijnden, 2023; Albergo et al., 2023), diffusion models (Sohl-Dickstein et al., 2015; Song & Ermon, 2020; Song et al., 2021), and Schrödinger bridges (Wang et al., 2021; Liu et al., 2022; 2024a; Shi et al., 2023; Gushchin et al., 2023b;a), and (Somnath et al., 2023) are deeply interconnected under the framework of generalized bridge matching (Tong et al., 2023; Albergo et al., 2023; Tong et al., 2024a; Shi et al., 2024). Recently, there has also been attempts to understand diffusion models as approaches to minimize the dynamic Wasserstein distances (Kwon et al., 2022; Khrulkov et al., 2023). Another recent work extends the flow matching to the flows on Riemannian manifolds (Chen & Lipman, 2024; Atanackovic et al., 2025). Recent works generalize flow-matching from different perspectives, Chen & Lipman (2024) generalize the flow-matching to the flows on Riemannian manifolds, Atanackovic et al. (2025) attempt to extend the flow-models to return meaningful flows for the data beyond training distributions, and Haviv et al. (2025) generalize the flow matching to the cases where data can be treated as distributions of distributions.

Additionally, there has been recent dynamic extension to the conditional neural optimal transport (Hosseini et al., 2023; Kerrigan et al., 2024). There has also been efforts to study neural network based scalable approaches to solve high-dimensional partial differential equations (Wan et al., 2023).

# D Implementation Details

## D.1 EMNIST Classifier

We merged the whole alphabet into one class and each number is treated as a separate class (digits between 0 and 9 are given same label as their value and any letter is labeled 10). In order to circumvent the effects of data imbalancedness on classifier training, we employed the class-reweighted softmax loss function. For $k$-class classification, consider the vector $\boldsymbol{z} \in \mathbb{R}^k$ containing the counts for class in the training data, we define the reweighting vector $\boldsymbol{\omega} \in \mathbb{R}^k$ with

$$\omega_i = \left( \sum_{j=1}^N \frac{z_i}{z_j} \right)^{-1}, \, \forall \, i \in [k]. \tag{52}$$

For one hot encoded label vector $\boldsymbol{y}$ and softmax activation output at neural network output $\hat{\boldsymbol{y}}$, the reweighted loss (risk) is given by

$$\ell(\boldsymbol{y}, \hat{\boldsymbol{y}}) = \mathbf{1}_k^\top (\boldsymbol{\omega} \odot \boldsymbol{y} \odot \hat{\boldsymbol{y}}) \tag{53}$$

The classifier for EMNIST is trained with the same train/validation split as provided in EMNIST dataset (Cohen et al., 2017). We trained the classifier with ResNet-18 (He et al., 2016) architecture and class-reweighted softmax loss function in equation 53. Adam optimizer (Kingma & Ba, 2014) along with warmup-cosine learning rate scheduler (Loshchilov & Hutter, 2017) is used to train the classifier with peak learning rate of $1 \times 10^{-3}$ with 500 warm-up steps. Total decay steps for cosine scheduler are set to 20,000 with end-value of learning rate set to be equal to $1 \times 10^{-5}$. The classifier training is stopped after 20,000 training steps, when classifier achieves more than 90% overall validation accuracy and 99% accuracy on digits. Confusion matrix of classifier are given in Appendix E.

### D.2 MNIST-EMNIST TRANSLATION MODELS

For the static domain translation, the transport network $T$ is a U-Net Ronneberger et al. (2015) with base-factor of 48 and the critic network $\eta$ is ResNet-51 He et al. (2016). In order to train both transport and critic networks, Adam optimizer Kingma & Ba (2014) is used with initial learning rate of $1 \times 10^{-4}$, which is scheduled to be halved after $10,000 + 5000c$, $20,000 + 5000c$, $30,000 + 5000c$, $40,000 + 5000c$ and $70,000 + 5000c$ training steps. Algorithm 1 is used for training with $50,000$ *learning iterations* with 10 *T update steps* for each $\eta$ *update step*, our training settings for static case are very similar to those of Gazdieva et al. (2023). For dynamic subset selection, following the settings from Neklyudov et al. (2023), the vector field $\varphi_t$ is parametrized using a U-Net with time embeddings from DDPM (Song & Ermon, 2020). Similar to action matching (Neklyudov et al., 2023), $\varphi_t$ is parametrized to return scalar by $\varphi_t(\boldsymbol{x}) = \langle \text{U-Net}(x), x \rangle$. Likewise, $Q_t$, which parametrizes $\rho_t$, is also a U-Net with time embeddings. We used AdamW optimizer with learning rate scheduling for 50,000 iterations. The optimizer parameters are $\beta = (0, 0.999)$, weight decay $= 0.1$ and drop out $= 0.1$. Additionally, we also employed exponential moving averages (EMA) in the training with the ema-rate 0.999. These settings are very similar to rectified flow matching and action matching (Liu et al., 2023; Neklyudov et al., 2023). Learning rate linearly increases from 0 to maximum value during first 5,000 iterations and then stays constant at maximum value with maximum learning rates of $2 \times 10^{-4}$ and $1 \times 10^{-4}$ for $\varphi_t$ and $Q_t$, respectively. Additionally, we clipped gradients to lie within [-1, 1]. Algorithm 2 is employed with 50,000 training iterations and 2 $\varphi_t$ for each $\rho_t$ update.

### D.3 MODELS FOR PU-LEARNING USING SUBSET ALIGNMENT

For PU learning with both static and dynamic subset alignment based approaches respectively, model architectures are given in code listings D.3 and D.3, respectively. For all models `num_hid` is set to be 1024, for `Smodel` and `etamodel`, the parameter `num_out` is by definition 1, whereas for `Qmodel` and `Tmodel`, outputs are set to be equal to data dimension. For both static and dynamic models, we used Adam optimizer Kingma & Ba (2014), with default settings, and learning rates $1 \times 10^{-4}$ and $2 \times 10^{-5}$ respectively. Additionally, we used EMA with ema-rate of 0.999 to evaluate models on both the test dataset and the validation datasets. We trained the model for the total of 20,000 *learning iterations*, with 10 *T update steps* for single $\eta$ *update step* using the Algorithm 1. Similarly, Algorithm 2 is employed to train neural networks for dynamic subset alignment. *learning iterations*, with 2 $\varphi_t$ *update steps* for single $\rho_t$ *update step*. The dynamic models contain time embeddings with trainable parameters. We employed the Adam algorithm for gradient based updates of neural network parameters. For all the tests for PU learning we fix $c = \frac{1}{\pi_+}$. For each data set same batch sizes are used to train both static and dynamic models and table 5, gives the values.

| Dataset | n | dim | $\pi$ | batch size |
|---|---|---|---|---|
| Abalone | 4177 | 8 | 0.16 | 20 |
| Banknote | 1372 | 4 | 0.44 | 10 |
| Breast-w | 699 | 9 | 0.34 | 10 |
| Diabetes | 768 | 8 | 0.35 | 6 |
| Haberman | 306 | 3 | 0.26 | 6 |
| Heart | 270 | 13 | 0.44 | 6 |
| Ionosphere | 351 | 34 | 0.64 | 6 |
| Isolet | 7797 | 617 | 0.04 | 4 |
| Jm1 | 10885 | 21 | 0.19 | 20 |
| Kc1 | 2109 | 21 | 0.15 | 20 |
| Madelon | 2600 | 500 | 0.5 | 20 |
| Musk | 6598 | 166 | 0.15 | 20 |
| Segment | 2310 | 19 | 0.14 | 20 |
| Semeion | 1593 | 256 | 0.1 | 4 |
| Sonar | 208 | 60 | 0.53 | 4 |
| Spambase | 4601 | 57 | 0.39 | 20 |
| Vehicle | 846 | 18 | 0.26 | 6 |
| Waveform | 5000 | 40 | 0.34 | 20 |
| Wdbc | 569 | 30 | 0.37 | 6 |
| Yeast | 1484 | 8 | 0.31 | 10 |

Table 5: UCI datasetets for PU Learning, along with total number of data points (n), dimension (dim), positive prior ($\pi$) and batch sizes employed in training the correspsonding models.

```python
import jax
from jax import numpy as jnp
from flax import linen as nn
import math
'''
etamodel: neural network parameterization for eta function
Tmodel: neural network parameterization for T function
'''
class etamodel(nn.Module):
  num_hid : int
  num_out : int
  @nn.compact
  def __call__(self, x):
    h = nn.Dense(self.num_hid)(x)
    h = nn.swish(h)
    h = nn.Dense(self.num_hid)(h)
    h = nn.swish(h)
    h = nn.Dense(self.num_hid)(h)
    h = nn.swish(h)
    h = nn.Dense(self.num_out)(h)
    return h

class Tmodel(nn.Module):
  num_hid : int
  num_out : int
  @nn.compact
  def __call__(self, x):
    def transport_net(x):
        MLP_out = nn.Sequential([
          nn.Dense(self.num_hid),
          nn.swish,
          nn.Dense(self.num_hid),
          nn.swish,
          nn.Dense(self.num_hid),
          nn.swish,
          nn.Dense(self.num_hid),
          nn.swish,
          nn.Dense(self.num_out),])(x)
```

```
39          ResConnect = nn.Dense(self.num_out)(x)
40          return MLP_out + ResConnect
41      output = transport_net(x)
42      return output
```

Listing 1: Model architectures for PU-Learning with static subset alignment

```python
import jax
from jax import numpy as jnp
from flax import linen as nn
import math

class Smodel(nn.Module):
  num_hid : int
  num_out : int

  @nn.compact
  def __call__(self, t, x):
    if jnp.ndim(t) == 0:
        t = jnp.broadcast_to(t, x.shape[0:-1]+(1,))
    h = jnp.concatenate([t,x], axis=-1)
    h = nn.Dense(self.num_hid)(h)
    h = nn.swish(h)
    h = nn.Dense(self.num_hid)(h)
    h = nn.swish(h)
    h = nn.Dense(self.num_hid)(h)
    h = nn.swish(h)
    h = nn.Dense(self.num_hid)(h)
    h = nn.swish(h)
    h = nn.Dense(self.num_out)(h)
    return h

class Qmodel(nn.Module):
  num_hid : int
  num_out : int

  @nn.compact
  def __call__(self, t, x_0, x_1):

    h = jnp.concatenate([t, x_0, x_1, t<0.5], axis=-1)
    h = nn.Dense(self.num_hid)(h)
    h = nn.swish(h)
    h = nn.Dense(self.num_hid)(h)
    h = nn.swish(h)
    h = nn.Dense(self.num_hid)(h)
    h = nn.swish(h)
    h = nn.Dense(self.num_hid)(h)
    h = nn.swish(h)
    h = nn.Dense(self.num_out)(h)

    x_t = (1-t)*x_0 + t*(x_1) + t*(1-t)*h

    return x_t
```

Listing 2: Model architectures for PU learning with dynamic subset alignment

### D.4 IMAGE-TO-IMAGE TRANSLATION ON FFHQ

In our experiments for static subset alignment, we used a three layered MLP architecture with swish activation functions in hidden layers to parameterize both the transportation map $T$ and the potential $\eta$. For the network parameterizing $T$, an additional skip connection connecting input and output is also used, which also contains a linear mapping, without any non-linear activation. Dimension of hidden layers are set to 1,024 for both Networks. Output dimension of the transport network is same as its input dimension (512), whereas potential network returns a scalar output. The Adam optimization algorithm is used to train both networks with a fixed learning rate of $1 \times 10^{-5}$. We employ EMA with ema-rate 0.999 in the training process. Algorithm 1 is used in the training with 50,000 *learning iterations* with 5 *T updates* for each $\eta$ *update*.

In order to train the dynamic models, the model architectures employed are also three layered MLPs but with time embeddings. The neural network parameterizing $\varphi_t$ is a three layers MLP with 64 dimensional time embeddings, 1,024 dimensional hidden layers, and a scalar output. The neural Network parameterizing $\rho_t$ contains two branches for static and dynamic components respectively. The dynamic part of network parameterizing $\rho_t$ also contains 64 dimensional time embeddings. We also use EMA with ema-rate 0.999 to train both networks, and a fixed learing rate of $1 \times 10^{-5}$. Dynamic models are trained using algorithm 2 for 50,000 *learning ierations* with 1 $\varphi_t$ *update* for 5 $\rho_t$ *updates*.

# E   CONFUSION MATRICES FOR MNIST → EMNIST DOMAIN TRANSLATION

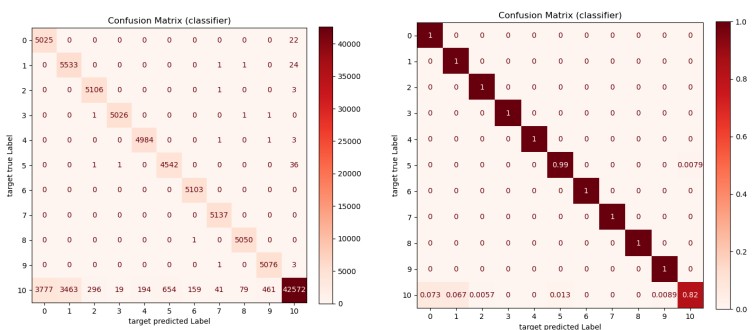

(a) Unnormalized confusion matrix        (b) Normalized confusion matrix

Figure 6: Confusion matrices for EMNIST classifier discussed in section 4.1

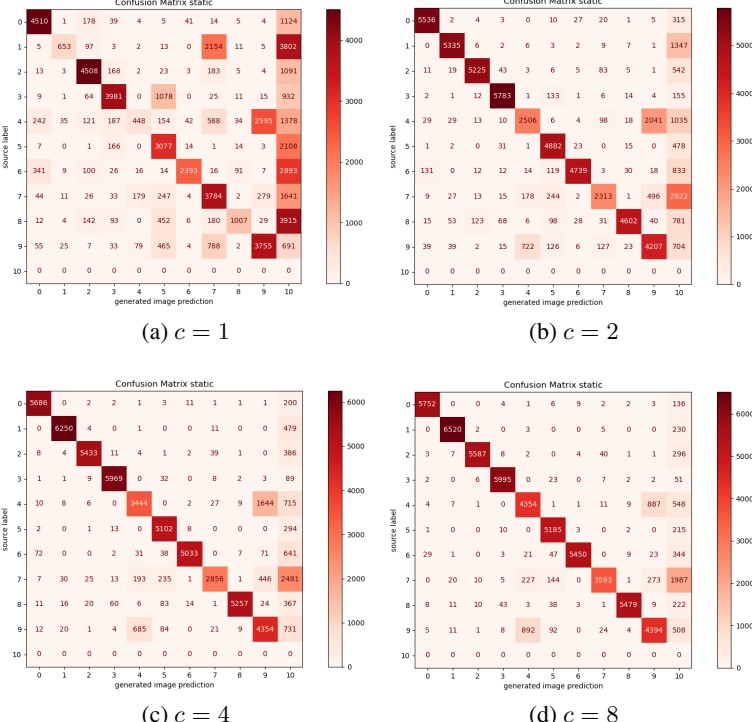

(a) $c = 1$        (b) $c = 2$

(c) $c = 4$        (d) $c = 8$

Figure 7: Confusion matrices for MNIST→EMNIST domain translation using static subset selection. Accuracy is computed by computing ratio between trace and some of all entries of confusion matrices.

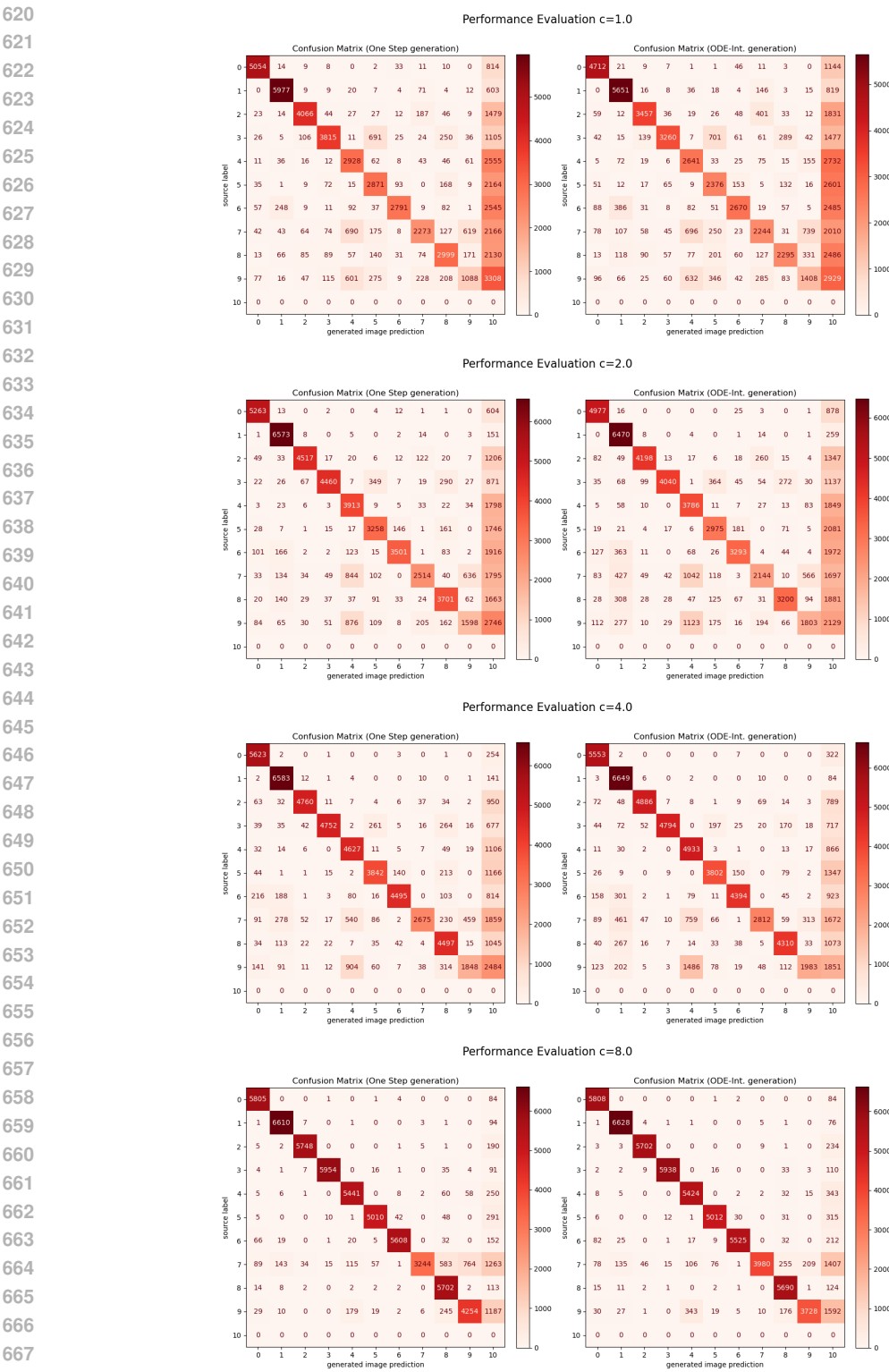

Figure 8: Confusion matrices for MNIST→EMNIST domain translation using dynamic subset selection.

# F    RESULTS FROM FFHQ

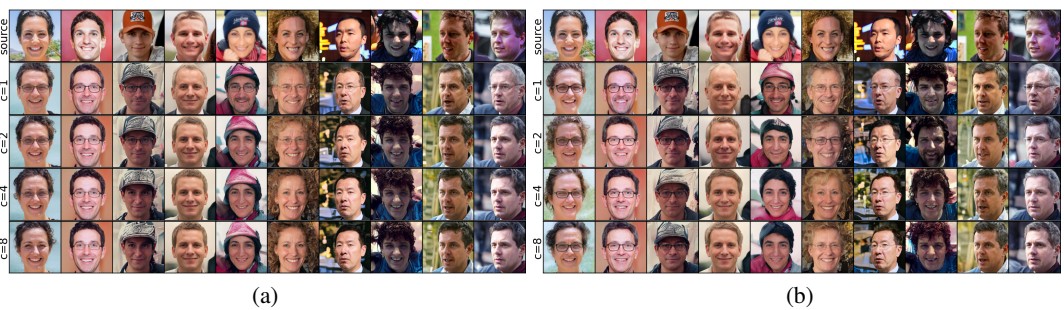

(a)                                                    (b)

Figure 9: FFHQ young→old translation using (a) static and (b) dynamic subset selection. Dynamic subset selection. Dynamic subset selection is evaluated using Euler integration with 100 steps.

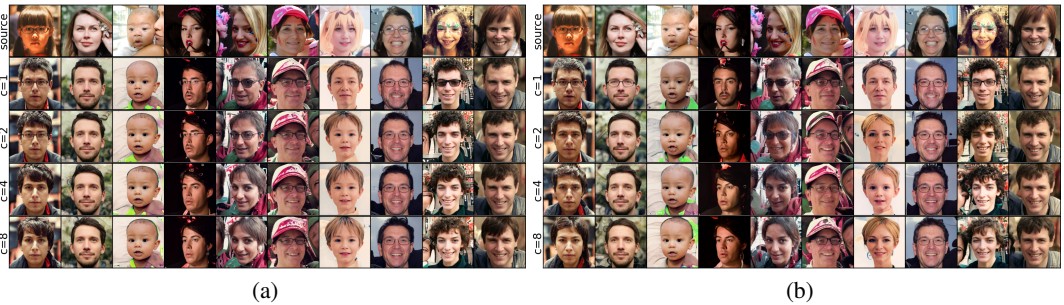

(a)                                                    (b)

Figure 10: FFHQ woman→man translation using (a) static and (b) dynamic subset selection. Dynamic subset selection is evaluated using Euler integration with 100 steps.

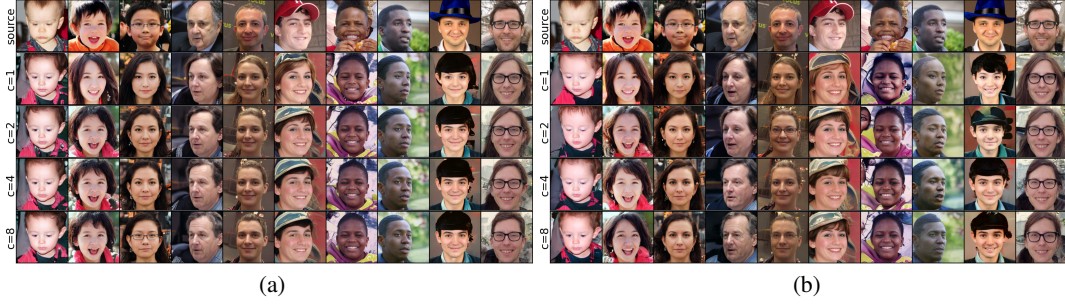

(a)                                                    (b)

Figure 11: FFHQ man→woman translation using (a) static and (b) dynamic subset selection. Dynamic subset selection is evaluated using Euler integration with 100 steps.

