# OpenReview forum: "Neural Optimal Transport for Subset Alignment"
_ICLR.cc/2026/Conference — ICLR 2026 Conference Withdrawn Submission_

### Official Review · Reviewer_xCZv · 2025-10-30

**Soundness:** 2
**Presentation:** 2
**Contribution:** 2
**Rating:** 2
**Confidence:** 4

**Summary:**

The paper proposes a static and a dynamic neural optimal transport solver for subset alignment between distributions with different masses. The goal is to match the source distribution to only a subset of the target. The authors consider the static and dynamic formulations of subset-selection for optimal transport which actually corresponds to semi-partial OT formulations presented as particular instances of semi-unbalanced OT ones. Here, the authors use the convex indicator function as particular $f$-divergence terms which enforce the partial matching constraint. For static formulation, the paper proposes a max–min objective that admits gradient-based optimization.

The second formulation is dynamic, extending the Benamou--Brenier formulation of the Wasserstein-2 problem. To avoid explicit time integration, the authors adopt a flow-matching-like approach using a time-dependent neural network $Q_t(x_0, x_1)$ for interpolation. For subset alignment, they alternate forward and backward integration, guided by samples from a learned subset of the target to ensure consistent mass transport across both flow directions.

Both formulations are evaluated on the MNIST–EMNIST image-to-image translation task and on unbalanced classes from ALAE. The reported metrics measure how accurately each method matches the target distribution. The authors also explore the Positive Unlabeled problem.

**Strengths:**

**[S1]** Both approaches, the static and the dynamic, allow varying $c$, enabling control over the proportion between the original target distribution and the reweighted set.

**[S2]** Overall, the flow of the paper is good and rather easy to understand.

**Weaknesses:**

I have two main concerns (W1, W2):

**[W1]** First, I think that the contribution of the paper for the static subset selection is not clear. This problem is closely related to unbalanced and partial Optimal Transport, which has already been extensively studied in the literature (Yang, 2018; Lübeck, 2022; Choi, 2024; Eyring, 2024; Klein, 2023). In fact, the proposed static formulation seems to have been already introduced and studied  in (Gazdieva, 2023), making the novelty of the static component unclear.

**[W2]** While the proposed dynamic approach is more interesting and implements some novel ideas; it lacks some theoretical validations. More precisely, as part of the dynamic problem, the authors have to deal with the time integration part from (8), which is not trivial for $c>1$ (what actually is the core part of the subset selection problem). To sort this out, the authors propose a pipeline that introduces sampling from a mixture of the distributions $\nu_{\varphi_1}$ and $\nu$; however, the validity of the interpolation introduced in Lines 224–225 is not properly justified and appears to rely on a heuristic.

Additionally, I have concerns regarding the experimental section.

**[W3]** In addition to my previous comment in [W2], I believe the experimental section should include a toy setup to better demonstrate the performance and practical validity of the proposed dynamic subset selection approach in subsection 2.2.  This is important given that the main contribution of the paper lies in the proposed dynamic integration method and sampling strategy. Moreover, the paper lacks a proper discussion on why the proposed dynamic method does not outperform state-of-the-art approaches (as observed in the FFHQ setup in Table 4). Such an analysis would help clarify the method’s limitations and highlight possible directions for future improvement.

**[W4]** The experimental section for the FFHQ dataset does not include a comparison with the method by (Gazdieva, 2023), which seems to be directly equivalent to the static approach proposed in the paper.

**[W5]** Additionally, the results in Table 4, which are taken from (Gazdieva, 2024), are based on an unbalanced OT formulation rather than the semi-unbalanced formulation which is closer to the semi-partial formulation considered in the paper. This leads to an unfair comparison as the proposed method and the methods under comparison solve different problems (and solving the unbalanced problem is typically more difficult that its semi-unbalanced counterpart).

**[W6]** Another concern is the performance of the proposed static and dynamic methodologies, as they do not outperform existing approaches on the FFHQ experiment setup. For example, in Table 4, target accuracy values are beaten by the UOT-SD method, while the class accuracy is mostly beaten by U-LOT.

**[W7]** The MNIST–EMNIST experiment offers limited insight, as the digits in both datasets are almost identical, making it difficult to evaluate the method’s performance. A more meaningful setup can involve a colored MNIST variant with disjoint source and target subsets (e.g., different digits). Moreover, the visual quality in Figure 2 appears inconsistent with the accuracy reported in Table 1, for example the one step integration approach has poor visual quality, but this method beats all the other approaches according to the reported accuracy. This raises doubts about the usefulness of this experiment and the reported metric.

**Minor Comments**

- The experimental section should highlight the best results in Table 4 for clarity.
- The contributions paragraph should be more clearly structured, as the current version is difficult to follow and does not clearly distinguish between theoretical and practical contributions. I suggest presenting the contributions in a list format, which is common practice and improves readability.
- I think that the paper should more properly introduce the intuition behind constant $c$ as it plays an important role in the performance of the methods.
- In line 073-074 the sentence: “Recent work on neural …” is incomplete as it does not include a verb to understand the idea.
- The range for $t$ should be specified.
- In equation 5, it should write $D_{i_{[a,b]}}$ instead of $D_{\varphi}$.
- It is necessary to specify that $\eta$ is a function as the notation may lead to confusion.
- Figure 1 is never referenced in the main text, this makes it difficult to properly interpret it.
- In section 2.2 it is necessary to add a reference for the Bernamou-Brenier formulation of Wasserstein-2 distance.
- The meaning of the word ‘judiciously’ in line 214 is not clear.
- In line 184, I think that the “for” before $\varphi_t$ is not necessary as it leads to confuse the role of $\varphi_t$.
- In line 338, the phrase starting with “PU learning…” is incomplete.
- Line 341, them -> the.
- Line 342, the word “applied” is repeated.
- Line 347, the word “learning” is repeated.

**To summarize**, I think the theoretical and practical contributions of the proposed static subset selection approach are not particularly novel, as comparable methods and findings have already been reported in earlier works. While the dynamic case introduces some interesting ideas, they lack a deeper theoretical and practical validation. My main concern is that the proposed integration methodology as well as the sampling strategy use some heuristic approach that is neither theoretically nor practically validated. Thus, I think that the paper needs a major revision and change its focus to better develop the proposed dynamic solution.

(Gazdieva, 2023) Milena Gazdieva, Alexander Korotin, Daniil Selikhanovych, and Evgeny Burnaev. Extremal domain translation with neural optimal transport. Advances in Neural Information Processing Systems, 2023.

(Gazdieva, 2024) Milena Gazdieva, Arip Asadulaev, Evgeny Burnaev, and Alexander Korotin. Light unbalanced optimal transport. In The Thirty-eighth Annual Conference on Neural Information Processing Systems, 2024.

(Yang, 2018) K. D. Yang and C. Uhler. Scalable unbalanced optimal transport using generative adversarial networks. In International Conference on Learning Representations, 2018.

(Lübeck, 2022) F. Lübeck, C. Bunne, G. Gut, J. S. del Castillo, L. Pelkmans, and D. Alvarez-Melis. Neural unbalanced optimal transport via cycle-consistent semi-couplings. arXiv preprint arXiv:2209.15621, 2022.

(Choi, 2024) J. Choi, J. Choi, and M. Kang. Generative modeling through the semi-dual formulation of unbalanced optimal transport. In Advances in Neural Information Processing Systems, volume 36, 2024.

(Klein, 2023) D. Klein, T. Uscidda, F. Theis, and M. Cuturi. Generative entropic neural optimal transport to map within and across spaces. arXiv preprint arXiv:2310.09254, 2023.

(Eyring, 202) L. Eyring, D. Klein, T. Uscidda, G. Palla, N. Kilbertus, Z. Akata, and F. Theis. Unbalancedness in neural monge maps improves unpaired domain translation. In The Twelfth International Conference on Learning Representations, 2024.

**Questions:**

- What is the connotation of ‘lags behind’ in line 212?
- In line 313, the authors state that one step integration performs worse in terms of visual perception; however, the accuracy in Table 1 shows the opposite. What is the reason for this behavior?
- Why was the swish activation function used?

---

### Official Review · Reviewer_VxVg · 2025-10-31

**Soundness:** 2
**Presentation:** 2
**Contribution:** 2
**Rating:** 2
**Confidence:** 3

**Summary:**

The paper proposes static and dynamic neural optimal transport (OT) formulations to address domain alignment where the source distribution should map to a subset of the target distribution. This is achieved by relaxing the target marginal constraint via a bounded density-ratio ($c \geq 1$). A learned potential function is then used to identify the target subset, ostensibly by its sign. The method is demonstrated on tasks including MNIST to EMNIST alignment, PU-learning, and latent-space translation for FFHQ attributes using a pre-trained ALAE model.

**Strengths:**

The paper addresses an important formal problem: aligning imbalanced domains or domains where the source only corresponds to part of the target. The authors provide both static and dynamic formulations, offering flexibility. The inclusion of practical training recipes is helpful. The paper is well-written and easy to follow. In general the method is novel.

**Weaknesses:**

The dynamic formulation of the proposed method does not seem useful; it was only added to extend the scope of the method. It only distracts from the main contribution of the paper.  The static formulation performs better on every dataset, except for the MNIST dataset, which is very simple, where it performs better according to the metrics. Visually, however, the performance of the dynamic formulation is questionable.

The success of the method hinges on the density-ratio bound c. Although the paper acknowledges the critical nature of this hyperparameter, it fails to provide a principled procedure for selecting it. This significantly hinders the method's practicality and reproducibility, as it appears to require expensive grid searching for each dataset.

Training complexity and reporting: The dynamic model introduces significant training complexity, for example, mixture sampling schedules and 100-step Euler integration. However, the paper provides no information on computational or memory costs. How does the computational cost compare to that of the Gaussian-based LOT method (Gazdieva et al.)?  Furthermore, the ablation in Figure 2 (MNIST to EMNIST) shows that 1-step integration performs poorly, suggesting that the 100-step integration is necessary, which exacerbates the cost concern.

In the FFHQ ALAE experiment, however, the 512-dimensional latent space is well structured, making it easy to find a map in this latent space. Doing some experiments with this dataset and the ALAE autoencoder, I know how easy it is to achieve well-looking results here with even with a complex 100-step ODE methods.  An obvious and informative baseline would be a shallow MLP regression with the same number of parameters as your method, mapping source latents to target latents.

**Questions:**

**Q1**:  How should a practitioner select the density-ratio bound $c$ in practice, absent an exhaustive grid search? Have you explored learning $c$ as a parameter, or adapting it via a schedule or validation-driven tuning?

**Q2**: Can you provide evidence for the reliability of the potential's sign as a selector? Please provide ablations showing its robustness to the choice of $c$, dataset size, and training stability. What is the theoretical or empirical justification for the "bottom $1/c$ fraction" heuristic, and how sensitive is the model to this fraction?

**Q3**: When is the dynamic formulation demonstrably better than the static one, and is its extra complexity justified? The MNIST$\to$EMNIST results (Fig. 2) suggest mixed results. Please provide clear ablations clarifying the trade-offs.

**Q4**: Please report wall-clock/GPU hours and peak memory usage for static vs. dynamic (1-step vs. 100-step integration) on a common benchmark (e.g., FFHQ).

---

### Official Review · Reviewer_USoJ · 2025-11-01

**Soundness:** 2
**Presentation:** 2
**Contribution:** 1
**Rating:** 2
**Confidence:** 5

**Summary:**

This paper proposes two approaches, static and dynamic, to solve a relaxed Monge formulation for domain translation tasks. Motivated by class imbalance between source and target datasets, the authors adopt a partial optimal transport (OT) formulation. The static approach is a particular instantiation of the semi-dual unbalanced OT framework of [1], whereas the dynamic approach is based directly on the action-matching formulation of [2]. The methods are evaluated on image-to-image translation and positive–unlabeled (PU) learning tasks.

**Strengths:**

- The paper is well written and clearly organized, making it easy to follow the methodology and experimental setup.

- The proposed methods are theoretically grounded, relying on established formulations of unbalanced transport and dynamical OT.

**Weaknesses:**

I have significant concerns about both the originality of the method and the experimental validation.

- **Lack of methodological novelty.** The static formulation appears to be a direct special case of the semi-dual unbalanced OT method in [1], differing only in the specific choice of divergence $f$. The dynamic formulation closely mirrors the action-matching framework in [2], with minimal conceptual or technical deviation. As a consequence, the methodological contribution of the paper appears incremental.

- **Limited strength of experimental results.** The empirical results do not convincingly support the effectiveness of the proposed methods. The dynamic formulation struggles to scale to PU learning, which may reflect inherent limitations in the scalability of action-matching-based approaches. The performance of static form is generally comparable to, or slightly worse than, NTC-MI, calling into question the practical advantages of the proposed framework.

- **Comparison with UOTM [1]** In Figure 3 and Table 4, the comparisons use UOT-SD as the baseline. However, UOT-SD is an improved variant of UOTM designed to match distributions more accurately. In contrast, UOTM allows for target mismatch and is more suitable for class-imbalanced scenarios. Therefore, a more appropriate and fair comparison would be against UOTM with different choices of divergence functions $f$ (e.g., KL, softplus, chi-square), rather than solely against UOT-SD [3].

References

[1] Generative Modeling through the Semi-dual Formulation of Unbalanced Optimal Transport.

[2] Action Matching: Learning Stochastic Dynamics from Samples.

[3] Analyzing and Improving Optimal-Transport-based Adversarial Networks.

**Questions:**

Using the same experimental setup, could the authors run UOTM [1] with different divergence functions $f$ (e.g., softplus, KL)? How does the performance compare to the proposed static formulation?

---

### Official Review · Reviewer_vV9R · 2025-11-03

**Soundness:** 2
**Presentation:** 2
**Contribution:** 1
**Rating:** 2
**Confidence:** 4

**Summary:**

The paper proposes static and dynamic neural optimal transport (OT) formulations for subset alignment, where a source distribution is transported to a reweighted subset of a target distribution. The key relaxation enforces a bounded density ratio between the learned terminal distribution and the target distribution. This is expressed via an f‑divergence whose generator is the indicator over [0,c] (“range divergence”).

In the static case, the authors derive a dual objective in which a transport map T and a dual potential are parameterized by neural networks. In the dynamic case, they start from Benamou–Brenier and obtain a dual involving a time‑dependent potential plus a terminal penalty.

Experiments include MNIST-EMNIST domain translation (qualitative and accuracy using a digit/letter classifier), PU‑learning on 20 UCI datasets, and FFHQ latent‑space translations across gender and age.

**Strengths:**

1) the authors provided practical training details, concrete architectures and schedules are provided

2) the authors considered a dynamic version of the problem

**Weaknesses:**

1) The static algorithm seems to be completely identical to the incomplete OT algorithm from (Gazdieva, 2023). It is not clear what the difference is. The partial OT was simply rewritten as a special case of the unbalanced OT.

Extremal Domain Translation with Neural Optimal Transport (Gazdieva, 2023)
https://neurips.cc/virtual/2023/poster/70097

2) Why the dynamic version of the algorithm is needed is not clear. For example, judging by table 2, the dynamic version almost always loses to static version, which in fact already exists.

3) The experiments are all small-scale, that is, either on MNIST dataset or on some low-dimensional latent space. Neural network based OT methods for Extremal OT have been already fully scaled to work with images, so it is not clear why the proposed approach is needed then, since the authors did not demonstrate how it works for full-scale images.

4) Comparisons are only with unbalanced methods, there is no comparison for example with (Gazdieva, 2023), which solves the same partial problem.

5) Section 4.2 and appendix B contain the already well-known results that the Lagrange multiplier is zero in case of outliers. This was discussed in (Gazdieva, 2023). In addition, the formulation of the experiment itself is not very clear, and it could be described better in the main text.

6) The general comment is that the main text is not sufficiently clear, there are a lot of mathematical calculations with only minimal explanations + references to other articles. For example, the section 2.2 is written in a very dense style.

7) Empirical validation of the constraint itself. The paper does not measure whether the learned terminal distribution satisfies the intended density‑ratio bound (e.g., report the selected fraction and over‑weighting statistics vs. c)

8) Ablations and sensitivity: a) The mixture schedule for alpha^k in Eq. (11) is crucial for dynamic training but lacks ablation, b) The role of
c is shown qualitatively, but there is no automatic or data‑driven selection. A tuning protocol or cross‑validated criterion would help

9) Report variances over runs for FFHQ metrics.

10) In PU‑learning, the method underperforms on some datasets (e.g., Spambase, Waveform; Table 2). A short error analysis (feature dimension, class prior, sample size, etc.; Table 5) could clarify when the approach is preferable

**Questions:**

1) Constraint verification. Could you report empirical checks of the density‑ratio bound?

2) Mixture scheduler. How sensitive are results to the schedule alpha^k  in Eq. (11)?

3) Choosing c. Beyond the qualitative trends in Sections 4–5, can you propose a data‑driven selection rule (e.g., pick c minimizing a validation version of the objective, or using target‑class calibration for FFHQ)?

4) Failure modes in PU‑learning. For datasets where the method lags behind PUSB/NTC‑MI (Table 2), what patterns do you observe? Is performance correlated with dimension/sample size/class prior (see Table 5)?

5) Runtime and scalability. Can you report training time and memory vs. baselines for MNIST→EMNIST and FFHQ? Also, how does the dynamic ODE (100 steps) compare with one‑step generation in terms of cost and quality (Fig. 2)?

6) FFHQ metrics. Please provide mean±std over seeds for Table 4 and calibration analyses of the pretrained attribute classifiers used for evaluation. Are conclusions robust to a different attribute classifier?

---

### Note · Authors · 2025-12-04

**Comment:**

We are grateful to the reviewers for their through and comprehensive feedback for this paper. We would like to withdraw this submission and improve upon it for future submission.

**Withdrawal Confirmation:**

I have read and agree with the venue's withdrawal policy on behalf of myself and my co-authors.